



# The influence of irradiance and interspecific differences on $\delta^{11}B$, $\delta^{13}C$ and elemental ratios in four coralline algae complexes

Maxence Guillermic[1], Erik C. Krieger[2,3], Joyce Goh[1], Christopher E. Cornwall[2], Robert A. Eagle[1]

[1]Department of Atmospheric and Oceanic Sciences, Institute of the Environment and Sustainability, Center for Diverse Leadership in Science, University of California - Los Angeles, CA 90095

[2]School of Biological Sciences, Victoria University of Wellington, Wellington, New Zealand

[3]Red Sea Research Center, King Abdullah University of Science and Technology, Thuwal, Saudi Arabia

*Correspondence to*: Maxence Guillermic (maxence.guillermic@gmail.com) and Robert Eagle (robeagle@g.ucla.edu)

**Abstract.** Coralline algae are a cosmopolitan group of important foundational species. The calcium carbonate they produce is increasingly being investigated as paleoenvironmental archives, as well as used to trace physiological responses of these important macroalgae to environmental change. Here we address the impact of light (irradiance) on 4 species complexes of coralline red algae including two morphologies; geniculate (branching) and non-geniculate (encrusting). The four complexes up-regulated their $\delta^{11}B$ derived $pH_{CF}$ relative to seawater by 0.6 to 0.8 pH unit. $\delta^{11}B$ was not measurably affected by varing irradiance despite evidence of increasing photosynthesis constrained by measurements of photophysiological parameters and $\delta^{13}C_{mineral}$. All complexes were able to maintain and elevate their $pH_{CF}$ relative to seawater for all treatments. Non-geniculate and geniculate complexes had distinct geochemical signatures of $\delta^{11}B$, $\delta^{13}C_{mineral}$ and trace elements. These differences in geochemical signatures indicate a variety of calcification mechanisms exist within coralline algae.

We propose that different sources of dissolved inorganic carbon (DIC) are necessary to explain the observed $\delta^{13}C_{mineral}$. As geniculate species have higher photosynthetic activity (i.e. gross photosynthesis), the DIC sources allocated to calcification might be limited due to greater $CO_2$ drawdown. This is supported by B/Ca and U/Ca ratios suggesting modulation of carbonate chemistry and especially lower $DIC_{CF}$ in geniculate relative to non-geniculate complexes. DIC sources might come from direct $CO_2$ diffusion or better recycling of metabolic $CO_2$ which would explain the depleted $\delta^{13}C_{mineral}$. This strategy likely arises from the different energy needs of the organisms, with non-geniculate using relatively more energy to support calcification. We suggest the different calcification mechanisms between morphologies are linked to distinct photosynthetic activity strategies. While photosynthesis can provide energy to geniculate complexes to maintain their metabolic needs, their calcification may be limited by DIC. In contrast, non-geniculate forms, may benefit from more limited DIC drawdown due to lower photosynthetic activity, therefore maintaining higher internal DIC concentrations ultimately supporting faster calcification.



## 1 Introduction

Coralline algae are widespread foundational species found around the globe, and in some locations their calcium carbonate forms maerl or rhodolith beds which are the dominant benthic substrate of the area (Steneck et al., 1986). In other cases they can form ecologically and structurally significant contributions to other benthic environments, for example in tropical coral reefs (Cornwall et al., 2023) and within kelp forests (Connell, 2003b, Irving et al., 2004). As with other marine calcifiers, they are potentially threatened by ocean warming and acidification, but with some evidence they have plasticity and resilience to some of these climate change stressors (Anthony et al., 2008; Martin et al., 2013a; Cornwall et al., 2019).

Coralline algae are important in the field of paleoenvironmental reconstruction, particularly as they may grow in cooler regions such as the Arctic where other commonly used archives such as mounding corals or foraminifera are not available (e.g., Halfar et al., 2000; Kamenos et al., 2008; Anagnostou et al., 2019). Before using coralline algae carbonate as archives for paleoclimate reconstruction, a good understanding of biomineralization mechanisms and how those mechanisms are impacted by environmental stressors are needed. Carbonate skeletal $\delta^{11}$B has been used to explore internal pH (pH$_{CF}$) and carbonate chemistry regulation in coralline algae in response to environmental change such as ocean acidification (Cornwall et al., 2017; Donald et al., 2017; Sutton et al., 2018), with evidence suggesting that the calcifying environment of coralline algae have pH elevated with respect to seawater (Cornwall et al., 2017; Donald et al., 2017; Sutton et al., 2018) as has been observed in scleractinian corals (McCulloch et al., 2017; Eagle et al., 2022).

The most significant body of work on geochemical tracers of internal pH and carbonate chemistry regulation has primarily focused on symbiont bearing surface corals indicating that the photophysiology of the symbiont may influence the chemical regulation of calcification. For example, regulation of the pH of the calcifying medium within the calicoblastic epithelium is known to show day-night cycles (Al-Horani et al., 2003; Guillermic et al., 2021; Cameron et al., 2022). Corals that lose symbionts during temperature stress, may also exhibit a deregulation of the calcification fluid chemistry and anomalous skeletal geochemical signatures (e.g., D'Olivio et al., 2017; Guillermic et al., 2021; Cameron et al., 2022). Conversely, heat resilient corals may not undergo this process (Eagle et al., 2022). Varying light levels can also influence coral skeletal geochemistry in controlled culture experiments (Dissard et al., 2012; Juillet-Leclerc et al., 2014).

Coralline algae are photosynthetic organisms that inhabit various habitats where light fluctuates greatly. Increasing irradiance generally enhances calcification of coralline red algae (Goreau, 1963; Borowitzka 1981; Borowitzka and Larkum 1987; Martin et al. 2013a; Korbee et al., 2014; Egilsdottir et al., 2016; Krieger et al. 2023). Light reduction in high-light adapted species *Porolithon onkodes* had a significant negative impact on calcification and induced mortality (Bessell-Browne et al., 2017). Increasing irradiance on low-light adapted species resulted in photoinhibition (Kain, 1987; Sagert et al., 1997; Kühl et al., 2001; Roberts et al., 2002; Martin et al., 2013b). Although light clearly affects calcification, responses vary among studies and the mechanisms linking light, photophysiology, and calcification are still not fully understood.

Because calcification is active in the meristematic region where there is a high concentration of chloroplasts, a direct link between photosynthesis and the calcification space is hypothesised. Photosynthesis has multiple ways in which it could





promote calcification: 1) increase pH within the diffusive boundary layer surrounding the cells during the day via $CO_2$ removal, 2) provide the cell wall polysaccharides and proteins, and 3) provide energy to the cell formation and calcifying medium carbonate chemistry regulation (McCoy et al. 2023). Environmental parameters influencing irradiance in natural settings can change population communities and functionality of the ecosystem thus a good understanding of the mechanisms influencing calcification (including light) is needed to foresee changes due to future environmental challenges.

70        Krieger et al. (2023) explored the physiology and photophysiology of low-light CCA complexes *Phymatolithopsis repanda, Pneophyllum* spp. *Corallina* spp., and *Arthrocardia* spp. cultured under changing irradiances and proposed that light-enhanced calcification is the result of an elevated diffusion boundary layer pH which raises calcifying fluid $pH_{CF}$ and that $[Ca]_{CF}$ could be the limiting parameters for fast growing species as also observed in Comeau et al. (2019). Here we explore the underlying mechanisms behind interspecific differences and the effect of changing irradiance on coralline red algae

complex calcification using geochemical tracers, namely the boron, carbon and oxygen isotopic compositions ($\delta^{11}B$, $\delta^{13}C$, $\delta^{18}O$) as well as minor elemental compositions (Mg/Ca, Sr/Ca, Li/Ca, B/Ca, Ba/Ca).

## 2 Materials and Method

### 2.1 Specimens and culture experiment

        Culturing experiments on non-geniculate coralline algae of different morphology ("thick" = *Phymatolithopsis*

*repanda*; "smooth" = *Pneophyllum* spp.) as well as two groups of geniculate corallines ("fine" = *Corallina* spp.; and "robust" = *Arthrocardia* spp.) were described in a previous study (Krieger et al., 2023) an shown in Fig. 1. To briefly summarize this work, specimens were collected by scuba divers at depths between 1 and 2 m from two field sites located in Te Moana-o-Raukawa Cook Strait, Te Whanganui a Tara Wellington, Aotearoa New Zealand. Taxonomic and DNA-based identifications are described in Krieger et al. (2023). As mentioned in the latter paper, samples can form a complex containing

multiple species with a dominant presence of one species. Those complexes present characteristic physiological and geochemical responses. For clarity, we will further refer to the species in the text as *Phymatolithopsis* complex, *Pneophyllum* complex (both non-geniculate) and *Corallina/Arthrocardia* fine, *Corallina/Arthrocardia* robust (both geniculate) species. Specifically, *Phymatolithopsis* complex consists of *Phymatolithopsis repanda* (*Hapalidiales* ZT 75% and *Hapalidiales* sp. D 25%). *Pneophyllum* complex consists of *75% Pneophyllum* sp. F and 25% *Corallinales* sp. E. *Corallina/Arthrocardia*

morphologies fine and robust consists of 75% *Corallina* sp. and 25% *Arthrocardia* sp.

        The original culture experiment was conducted over the 2019 summer and autumn (17th February to 19th May) in the facilities of the Victoria University of Wellington Coastal Ecology Laboratory. A detailed description of the original tank experiment can be found in Krieger et al. (2023) but we will briefly outline the most important information relevant for the present study here. The study organisms were exposed for 85 days to four different light levels (daily doses 0.6, 1.2, 1.8,

2.3 mol photons $m^{-2} d^{-1}$; noon peak irradiance 20, 40, 60, 80 µmol photons $m^{-2} s^{-1}$) that represent naturally occurring



subcanopy irradiances at the collection sites. Each irradiance level (i.e. treatment) was replicated twelve times on the tank level. The twelve tanks from each treatment were distributed over eight water baths with each bath housing between one to two tanks from each treatment. Header tanks supplied six random tanks which were equally distributed between two neighboring water baths with 150 mL min⁻¹ of fresh filtered (10 µm) seawater each. Water bath and header tank identity of

each experimental tank was later used during the statistical analysis to remove sample interdependence. Light was provided by LED panels which simulated a natural diel light cycle and mimicked a coastal underwater light spectrum. Temperature control was achieved by using submersible heaters and aquarium chillers with the difference in mean treatment temperature between treatments was not higher than 0.1 °C (highest $16.45 \pm 0.1$ SE and lowest $16.36 \pm 0.1$ SE). Seawater carbonate chemistry was monitored frequently through the measurement of tank $pH_T$ and total alkalinity. Mean treatment total alkalinity

was within 4 µmol.kg⁻¹ (highest $2279.77 \pm 3.41$ SE and lowest $2275.11 \pm 4.88$ SE) while $pH_T$ was within 0.1 units (highest $8.02 \pm 0.01$ SE and lowest $8.01 \pm 0.01$ SE).

## 2.2 Specimens and culture experiment

Photosynthetic (Chl *a* content, Fv/Fm, ETRmax, gross photosynthesis) and physiological (net calcification) parameters as well as tissue $\delta^{13}C$ were originally published in Krieger et al. (2023) and are also presented in Table S1.

Physiological data against irradiance are also presented in Fig. 2.

## 2.3 Carbonate geochemistry

Methods used in this study were previously described in Guillermic et al. (2020, 2021, 2022) and Eagle et al. (2022). Briefly, powdered calcium carbonate samples were organically cleaned using a solution of 0.2 % hydrogen peroxide. Samples were dissolved in 1 N HCl and purified for boron isotopes through microdistillation (Gaillardet et al., 2001, Wang et al., 2008).

Boron isotopic measurements were carried out on a Thermo Scientific® Neptune MC-ICP-MS at the Pôle Spectrométrie Océan (PSO), Plouzané and at the Dornsife PLASMA Facility of the University of Southern California, Los Angeles.

Elemental ratios were measured on a Thermo Fisher Scientific Element XR HR-ICP-MS at the PSO, Ifremer (Plouzané, France) after [Ca] analyses on an ICP-AES Ultima 2 HORIBA at the PSO (Plouzané, France). Data quality and external reproducibility were monitored by repeated measurement of JCp-1 (Gutjarh et al., 2021), NIST RM 8301 (Stewart et

al., 2020) and filtered seawater for both boron isotopes measurements and trace elements. $\delta^{11}B$ measured for NIST 8301 coral was $24.26 \pm 0.22$ ‰, 2 SE, n=19 (published value is $24.17 \pm 0.07$ ‰, 2 SE, n=7, Stewart et al., 2020), $\delta^{11}B$ of JCp-1 was $24.51 \pm 0.14$ ‰, 2 SE, n=12 (published value is $24.36 \pm 0.14$ ‰, 2 SE, n=10, Gutjarh et al., 2021) and $\delta^{11}B$ measured for a filtered seawater was $39.53 \pm 0.12$ ‰, 2 SE, n=2 (published value is $39.61 \pm 0.04$ ‰, 2 SE, n=28, Foster et al., 2010).

Analyses of carbonate skeletal $\delta^{13}C$ and $\delta^{18}O$ were carried out on a Matt 253 (Kiel IV carbonates, dual Inlet) mass

spectrometer at the stable isotope facility of Pôle spectrométrie Océan (PSO, Plouzané, France). Results were calibrated to the Vienna Pee Dee Belemnite (V-PDB) scale and referenced to the international standard NBS19.

Geochemical data analyzed in this study are presented in Table S1 and Fig. 3.



## 2.4 pH$_{CF}$ calculations

The pH$_{CF}$ was calculated from measurements of coral skeletal $\delta^{11}$B following Hemming and Hanson (1992) and equation from Zeebe and Galdrow, (2001):

$$pH_{CF} = pK_B^* - \log\left(-\frac{\delta^{11}B_{seawater} - \delta^{11}B_c}{\delta^{11}B_{seawater} - \alpha*\delta^{11}B_c - \varepsilon}\right) \qquad \text{eq. 1}$$

with pK$_B$*(T,S) representing the dissociation constant, temperature of 16.4 °C and salinity of 35 psu. $\delta^{11}$B$_{sw}$ is representing the boron isotopic composition of seawater (Foster et al., 2010), $\delta^{11}$B$_c$ representing the boron isotopic composition of the mineral (e.g. high-Mg calcite of coralline red algae), and $\varepsilon$ representing the boron isotopic fractionation between boric acid and borate ion (27.2 ‰, Klochko et al., 2006).

## 2.5 Statistical analyses

Linear and quadratic models were compared using Akaike information criterion (AIC) to determine which model described best the data (Figures S1, S2, Tables S2, S3). Only significant lines were plotted for the regressions that had a significant p-value (for linear fit) or R$^2$ (for quadratic fit) (Figures S1, S2). Statistical tests were performed between the geochemical data and matching photophysiological data from Krieger et al. (2023).

Normality of the data was assessed and data transformed using R to normalize the entire dataset (by variable) using Box-Cox transformation and then subsequently tested the normality of the data set using the Shapiro Normality Test and Q-Q plot.

ANOVA tests in R were used to evaluate the effect of irradiance and test differences between species. ANOVA tests that had a significant p-value were then further analyzed using the TukeyHSD Multiple Comparisons of Means test at a family-wise confidence level of 95%. Results are presented in Tables S4, S5, S6 and S7.

Mantel test is a statistical method that evaluates the correlation between multiple parameters and allows representation of complex datasets. Mantel tests were performed using R for each complex and are presented in Fig. 4.

Principal component analysis (PCA) was made using Graphpad Prism (version 10.2.3 for Windows GraphPad Software, Boston, Massachusetts USA, "www.graphpad.com") for all trace elements and physiological parameters. Relevant physiological parameters were selected, ETRmax and $\delta^{13}$C$_{organic}$ given the reduced amount of data (Figure 4).

The averages of photophysiological parameters presented in Figs 2, 4, and 6 are derived from the full dataset provided in the supplemental information of Krieger et al. (2023). Regression analyses and other statistical tests were conducted on a subset of photophysiological samples for which geochemical analyses were available (Table S1). Individual paired data and averages are shown in the cross-plots in Figs. 5 and 7 in order to display maximum information on the data.



## 3 Results

### 3.1 Net calcification and changing irradiance

No significant relationship was observed between net calcification and irradiance (p > 0.05, ANOVA) in our subset of data. Differences in net calcification were only significant between complexes (p < 0.05, ANOVA for irradiance 0.6, 1.8, 2.3). However, Krieger et al. (2023) presented two significant relationships, one non-linear for *Corallina* and one non-linear for *Spongites* when the full dataset was taken into account.

### 3.2 $\delta^{13}C_{mineral}$ and $\delta^{13}C_{tissue}$

The geniculate and non-geniculate complexes present different absolute values of $\delta^{13}C_{mineral}$ and responses with increasing irradiance. Relatively lower $\delta^{13}C_{mineral}$ values (~ -5.5 ‰) are observed for geniculate *Corallina/Arthrocardia fine* and *Corallina/Arthrocardia robust*; non-geniculate *Phymatolithopsis complex* and *Pneophyllum complex* have relatively enriched $\delta^{13}C_{mineral}$ signatures (~ -2.5 ‰). Significant differences in $\delta^{13}C_{mineral}$ between species were observed for all irradiances (Table S6).

ANOVA results indicate a significant effect of irradiance on $\delta^{13}C_{mineral}$ of the non-geniculate species (*Phymatolithopsis complex* and *Pneophyllum complex*) (p=0.01 and p=0.009, Table S4). These two complexes exhibit a significant linear increase in $\delta^{13}C_{mineral}$ with increasing irradiance levels (p=0.010 and p=0.003, respectively, Table S2). The geniculate *Corallina* is showing a non-linear ($R^2$=0.45) significant increase in $\delta^{13}C_{mineral}$ while *Corallina/Arthrocardia robust* is having a relatively stable $\delta^{13}C_{mineral}$ signature for the different treatments (p=0.948).

$\delta^{13}C_{tissue}$ data were already presented in Krieger et al. (2023). In our subset of samples, ANOVA supports a significant effect of irradiances for *Phymatolithopsis complex* and *Pneophyllum complex* (p=0.009, p=0.011). Values of $\delta^{13}C_{tissue}$ are linearly increasing with higher irradiances for *Phymatolithopsis complex* (p=0.001), and a significant non-linear relationship is observed for *Pneophyllum complex* ($R^2$=0.58). ANOVA also supports significant differences between species (Table S6). $\delta^{13}C_{mineral}$ are enriched in comparison to $\delta^{13}C_{tissue}$ by 9 to 22 ‰. Significant positive linear relationships between $\delta^{13}C_{mineral}$ and

$\delta^{13}C_{tissue}$ were observed for *Pneophyllum complex* and *Phymatolithopsis complex* (p=0.025, p=0.003), but not for *Corallina/Arthrocardia fine* and *Corallina/Arthrocardia robust,* Fig. 5A.

There is an increase in $\delta^{13}C_{mineral}$ with increasing net calcification across all complexes (p<0.001; Figure 5C). Some differences to note are that the geniculate *Corallina/Arthrocardia robust* and *Corallina/Arthrocardia fine* have the lightest $\delta^{13}C_{mineral}$ in line with observed lower net calcification. The non-geniculate complexes have higher net calcification and higher

$\delta^{13}C_{mineral}$, implying different sensitivities of net calcification to irradiance between complexes and difference between non-geniculate and geniculate complexes.



### 3.3 δ$^{11}$B

Enriched δ$^{11}$B values are observed for the geniculate *Corallina/Arthrocardia robust* (~26.4 ‰) and *Corallina/Arthrocardia fine* (~27.4 ‰), compared to the non-geniculate *Pneophyllum complex* (~24.5‰) and
*Phymatolithopsis complex* (~25.4 ‰). The differences between complexes are significant at irradiance 0.6, 1.8 and 2.3 (ANOVA p=0.008, p=0.001, p=0.006, respectively, Table S6). There are no significant linear relationships between δ$^{11}$B and irradiance (Tables S3 and S4).

No significant linear or non-linear regression was observed between δ$^{11}$B and irradiance. δ$^{11}$B differences were observed between species (ANOVA significant for most irradiances, Tables S3 and S4). T-tests show no significant differences
between *Corallina/Arthrocardia fine* and *Corallina/Arthrocardia robust* (geniculate) or *Phymatolithopsis complex* and *Pneophyllum complex* (non-geniculate) but do show significant differences between geniculate and non-geniculate species.

Crossplot of δ$^{13}$C$_{mineral}$ and δ$^{11}$B does show significant negative linear relationships across all complexes (p<0.0001), not significant at the complex level (Figure 5B). There is a clear distinction between non-geniculate and geniculate species. *Corallina/Arthrocardia robust* and *Corallina/Arthrocardia fine* show depleted δ$^{13}$C and high δ$^{11}$B while *Pneophyllum complex*
show enriched δ$^{13}$C and lower δ$^{11}$B (significant ANOVA).

δ$^{13}$C and δ$^{11}$B compared to net calcification and gross photosynthesis (Figure 5C, 5D, 5E and 5F) do not present any significant relationships. We note that higher δ$^{11}$B and lower δ$^{13}$C$_{mineral}$ coincides with higher gross photosynthesis and lower net calcification in the geniculate species while the opposite is true for non-geniculate species (Figure 5).

### 3.4 Trace elements

Li/Ca, B/Ca, Mg/Ca, Sr/Ca, Ba/Ca, U/Ca were analyzed in this study. Mg/Ca was the most impacted by irradiance between complexes, while Li/Ca was significantly impacted in *Pneophyllum complex* (p<0.001, ANOVA, Table S4) and Ba/Ca in *Corallina/Arthrocardia robust* (p<0.04, ANOVA). Most elements presented significant differences between complexes, including B/Ca, Li/Ca, Mg/Ca, Sr/Ca (ANOVA, Table S6).

Mg/Ca observed are significantly different between species at irradiance 0.6, 1.8 and 2.4 (p=0.047, p=0.03 and
p<0.001, ANOVA). Significant quadratic relationships between Mg/Ca and irradiance are observed for *Pneophyllum complex* and *Phymatolithopsis complex* (R$^2$=0.51, R$^2$=0.48) while a positive linear relationship is observed for *Corallina* (p=0.002) are best fit according to AIC analyses (Table S2, Figure S1). There is a significant impact of irradiance on Mg/Ca for *Corallina*, *Pneophyllum complex* and *Phymatolithopsis complex* (p=0.03, p=0.003 and p=0.04, ANOVA, Figure S1, Table S2, S4).

Significant positive relationships are observed between B/Ca and irradiance, quadratic for *Pneophyllum complex* and
linear for *Phymatolithopsis complex* (R$^2$=0.40, p=0.02 respectively) but not for other complexes. Based on TukeyHSD Multiple Comparisons of Means (see method section) B/Ca was significantly different for the species for the three irradiance treatments, 0.6, 1.2 and 1.8 (p=0.006, p=0.02 and p=0.0003 respectively, Figure S1, Tables S2, S6).



### 3.5 Other physiological parameters

ETR max (Maximum electron transport rate) is an important photophysiological parameter indicative of photosynthetic capacity and photosynthetic stress of photosynthetic organisms. ETR max is directly correlated to gross photosynthesis ($\mu gO_2/cm^2/h$) making it a key parameter to study the impact of changing irradiance in coralline red algae. In our subset of samples ETR max had significant positive linear relationships with irradiance for *Corallina/Arthrocardia robust*, *Corallina/Arthrocardia fine* and *Pneophyllum complex* (p= 0.035, p=0.0023, p=0.0238 respectively), Table S3, Fig. S2. Chl *a*, ETRmax and Fv/Fm were significantly different between species at different irradiance levels based on ANOVA (Table S6).

Significant differences between non-geniculate and geniculate complexes were observed in the photophysiological parameters. Net calcification was lower in geniculate complexes than in non-geniculate complexes (t-test, p<0.001). Gross photosynthesis was higher in geniculate complexes than in the non-geniculate ones (t-test, p<0.001).

### 3.6 Mantel test

Relationships between geochemical and physiological parameters are assessed using a Mantel test, as shown in Fig. 4.

### 3.7 Principal component analysis (PCA)

Principal component analysis (PCA) was performed for the geochemical and physiological data. The isotopic and trace element measurements were dissociated for better clarity of the figures. Vectors present a positive relationship between ETRmax and irradiance, a negative relationship between net calcification and $\delta^{11}B$, a positive relationship between net calcification and $\delta^{13}C_{mineral}$. There was a minor correlation but positive between irradiance and net calcification and between $\delta^{11}B$ and Fv/Fm (Figure 4).

In both cases, geniculate and non-geniculate species cluster together. Geniculate species *Corallina/Arthrocardia robust* and *Corallina/Arthrocardia fine*) show higher net calcification, higher $\delta^{13}C_{mineral}$ and lower $\delta^{11}B$. Non-geniculate species (*Pneophyllum complex* and *Phymatolithopsis complex*) on the contrary show lower net calcification, lower $\delta^{13}C_{mineral}$ and higher $\delta^{11}B$. The clustering is also observed with the trace elements. Geniculate species showing higher Li/Ca, Sr/Ca, Ba/Ca and U/Ca ratios than non-geniculate species.



## 4 Discussion

### 4.1 Trace elements


$\delta^{13}C$ of the mineral ($\delta^{13}C_{mineral}$) and the tissues ($\delta^{13}C_{tissue}$) can reflect photosynthesis and respiration (McConnaughey et al., 1997), where direct $HCO_3^-$ uptake from seawater enriches $^{13}C$ while recycling of respired $CO_2$ can decrease $\delta^{13}C$ of the DIC pool. Additionally, increased uptake of diffusive $CO_2$ (from seawater or metabolic) can result in depletion in $^{13}C$. Ultimately, the $\delta^{13}C_{mineral}$ reflects the relative abundance of photosynthetic $HCO_3^-$ uptake relative to respiration processes or

passive $CO_2$ diffusion from seawater. The $\delta^{13}C_{tissue}$ represents the source of DIC and kinetic fractionation by RUBISCO during photosynthesis, RUBISCO enzyme preferentially fixing $^{12}C$ leading to $\delta^{13}C_{tissue}$ being depleted relative to $\delta^{13}C_{mineral}$.

The positive relationships between $\delta^{13}C_{mineral}$ and irradiance, and $\delta^{13}C_{tissue}$ and irradiance in three out of four complexes highlights: 1) that irradiance impacts the geochemical signatures of the mineral, 2) photosynthetically driven isotope fractionation increases with increasing irradiance based on $\delta^{13}C_{mineral}$ and $\delta^{13}C_{tissue}$. Those results are in line with

photophysiological parameters measured (i.e. gross photosynthesis, ETRmax) showing increased photosynthesis with irradiance at the complex level.

Difference in sensitivities between $\delta^{13}C_{mineral}$ and irradiance is observed between *Pneophyllum complex* and *Phymatolithopsis complex* indicating complex-specific responses to light. In the range of irradiances tested in this study, geniculate complexes are less sensitive to changes in irradiance (p=0.975) than the non-geniculate ones (p=0.0001), Fig 5A.

There are clear differences in $\delta^{13}C_{mineral}$ signatures between non-geniculate and geniculate complexes. Non-geniculate complexes *Pneophyllum complex* and *Phymatolithopsis complex* are fast calcifiers that have enriched $\delta^{13}C_{mineral}$ and a strong response to increased irradiance. Geniculate complexes *Corallina/Arthrocardia fine* and *Corallina/Arthrocardia robust* present lower net calcification and lower $\delta^{13}C_{mineral}$. Those results are in line with net calcification being enhanced by photosynthesis (Figure 5C).

However, the story might be more complex when comparing the two morphologies. Direct photophysiological parameters (i.e. gross photosynthesis and ETR max) show that geniculate complexes have higher gross photosynthesis than the non-geniculate complexes, they also have lower $\delta^{13}C_{mineral}$ (Figure 5E). The higher photosynthesis rate in geniculate over non-geniculate has also been observed in the field (Nguyen et al., 2022). The discrepancy with $\delta^{13}C_{mineral}$ can be potentially explained if the source of DIC used by geniculate species is depleted in $^{13}C$. This could be the case if relatively more respired

$CO_2$ and/or direct diffusion from seawater contributes to the DIC pool than for non-geniculate complexes. The morphology of the geniculate represents a higher surface area-to-volume ratio and a thinner wall thickness; this might lead to greater passive transport of DIC to the site of calcification. On the contrary, the thick crust, and lower surface area to volume ratio of the non-geniculate species could lead to less passive diffusion as a source of DIC. Mao et al. (2024) established a carbon budget based on radiogenic-isotopes in the coralline red algae *Boreolithothamnion soriferum*. This budget highlighted that out of the 40 %

of the carbon used for photosynthesis, 15 % was recycled internally. We anticipate that this recycling will vary depending on morphologies and taxa and then impact $\delta^{13}C$. Bergstrom et al. (2020) showed various DIC uptake strategies depending on taxa,





especially $CO_2$ diffusion being more prevalent in basal taxa which highlight the diversity of carbon concentrating mechanisms in coralline algae.

From those results we show that the geochemical signatures of the mineral are impacted by changing irradiances
which allows us to investigate potential changes in $pH_{CF}$ constrained by boron isotopes.

## 4.2 Boron isotopes ($\delta^{11}B$)

The range of $\delta^{11}B$ for the complexes of this study has been observed for other coralline species *Neogoniolithum* sp.*, Sporolithum durum, Clathromorphum compactum* (Cornwall et al., 2018; Anagnostou et al., 2019; Sutton et al., 2018). We report $\delta^{11}B$ for four complexes of coralline red algae that showed average values ranging from 24.5 to 27.4 ‰, with significant
differences between species (ANOVA significant). This range of $\delta^{11}B$ seems consistent with sole incorporation of $B(OH)_4^-$ and realistic physiological modulation of $pH_{cf}$. However, we note that NMR study from Cusack et al. (2015) observed the presence of trigonal boron ($BO_3$) accounting for up to 30% of the total boron in *Lithothamnion glaciale*. The presence of $BO_3$ can also be due to the recoordination of $BO_4$ during the incorporation of boron within the crystal lattice (Klochko et al., 2009; Branson et al., 2015) which in that case would not impact the $\delta^{11}B$ proxy. NMR studies on other species of coralline red algae
along with boron isotopic measurements are lacking to affirm that $BO_3$ does not contribute to a part of the signal measured. For example, more extreme $\delta^{11}B$ data for *Neogoniolithon* were reported at ~31-40 per mill (Donald et al., 2017; Liu et al., 2020), even if $BO_3$ incorporation might not be the dominant driver, it could still contribute to the high values in that particular species/experiment (Donald et al., 2017; Liu et al., 2020).  In our study, the range of $\delta^{11}B$ reported (26 ± 3 ‰, 2 SD, n = 76, Figure 3B) is consistent with the pH at the site of calcification ($pH_{CF}$) and without further evidence of $BO_3$ incorporation and
impact on the $\delta^{11}B$, the $\delta^{11}B$ will be interpreted as a physiological signal in the following discussion.

## 4.3 $pH_{CF}$ is up-regulated relative to seawater

The primary calcification happens in the interfilament space in coralline red algae, secondary calcification happens within the cell walls (McCoy et al., 2023). It is thought that coralline red algae elevate their internal pH and modulate carbonate chemistry to promote calcification (Cornwall et al., 2017). $\delta^{11}B$ is thought to record the pH at the site of calcification ($pH_{CF}$).
Boron based studies suggest that $pH_{CF}$ is upregulated relative to seawater supporting favorable saturation state and calcium carbonate precipitation, as observed in corals (McCulloch et al., 2017; Cornwall et al., 2017; Anagnostou et al., 2019; Comeau et al., 2019) and other marine organisms (Sutton et al., 2018; Liu et al. 2020). The capacity of coralline algae to maintain its $pH_{CF}$ has also been shown to be impacted by ocean acidification, as recorded by the boron isotope proxy of pH at the site of calcification (Cornwall et al., 2017; Comeau et al., 2019) and indirectly seawater pH (Anagnostou et al. 2019).
As previously reported for other species, our data show upregulation of $pH_{CF}$ relative to seawater for the four complexes studied here with average values for *Corallina/Arthrocardia robust* and *Corallina/Arthrocardia fine* of 8.75 ± 0.21 (2 SD, n=19) and 8.81 ± 0.12 (2 SD, n=20), respectively and for *Pneophyllum complex* and *Phymatolithopsis complex* of 8.63 ± 0.20 (2 SD, n=18) and 8.68 ± 0.15 (2 SD, n=19), respectively (Figure 6). The seawater pH (total scale) during the experiment



was maintained to 8.02, meaning that internal pH for the four complexes was elevated relative to seawater by 0.6 to 0.8 pH
unit. Complex-specific $pH_{CF}$ are observed. The geniculate species (*Arthrocardia/Corallina* fine and robust) show higher $pH_{CF}$
in comparison to the non-geniculate complexes (*Pneophyllum complex* and *Phymatolithopsis complex*). All $pH_{CF}$ are in the
range to sustain the saturation state based on boron-based study in other marine organisms (McCulloch et al., 2017; Sutton et
al., 2018; Comeau et al., 2019; Liu et al., 2020; Guillermic et al., 2021 and others).

### 4.4 $pH_{CF}$ is not affected by changing irradiance at the complex level

As the first purpose of this study was to evaluate the impact of photosynthesis on $pH_{CF}$, we show that none of the four
low-light adapted complexes showed significant difference of $pH_{CF}$ for the different levels of irradiance. All complexes
presented pH homeostasis responses at different irradiance levels and despite evidence of increased photosynthesis activity
(Figure 6). These results highlight complex-specific $pH_{CF}$, the complexes are able to maintain an optimal $pH_{CF}$ demonstrating
a good acclimation in the range of irradiance tested (0.6 to 2.3 mol photon/m$^2$/day). This is also in line with the complexes not
showing significant changes in calcification with changing irradiances in our subset of samples (Table S3, Figures 2, S2). For
comparison, those $\delta^{11}B$-derived $pH_{CF}$ are higher than those measured via microelectrode in the light (8.15 - 8.30) in Arctic
corallines (Hoffman et al. 2018). This lack of response to changing irradiance may also result from photosynthesis-independent
mechanisms (de Beer and Larkum, 2001; Hofmann et al., 2016, 2018) helping to maintain favorable proton gradients.

### 4.5 Calcification space chemistry under changing irradiance

The response of calcification to photosynthesis is not fully understood in coralline red algae. While some studies
report a positive effect of photosynthesis on growth rate (Goreau 1963; Pentecost 1978; Comeau et al. 2014) others show non-
linear responses to increase irradiance (Martin et al. 2013b; Egilsdottir et al. 2016) or photoinhibition (Kain, 1987; Sagert et
al., 1997; Kühl et al., 2001; Roberts et al., 2002; Martin et al., 2013b). The subset of data we used for this study did not show
significant changes in net calcification, net calcification was maintained over the different treatments despite evidence of
increasing photosynthesis. In other words, photosynthetic activity was sufficient even at the lowest irradiance to 1) provide a
substantial provision of energy to the organism that can be allocated to active transports of ions and subsequent modulation of
the calcification space chemistry, 2) sustain a proton gradient between the calcifying space and seawater. This gradient is
maintained from elevation of pH surrounding the cells as result of photosynthetic rate and $CO_2$ drawdown by ions transporter
(Hoffman et al., 2016; Cornwall et al., 2013, 2014, 2017) and by the presence of light-mediated proton pump that is
independent from photosynthesis (Hoffman et al., 2016, 2018).

Increasing photosynthesis, however, can have other positive effects on the organism and calcification. For example
photosynthesis may sustain calcification by providing the key constituents of organic molecules needed for cell wall formation
which act as a template for mineral precipitation. Those organic molecules (like polysaccharides) can also have affinities with
Ca which can increase locally the saturation state and promote precipitation of $CaCO_3$. Overall, all complexes in this study
acclimatized well to the different levels of irradiance, calcification was maintained but not improved. This can also result from





other limiting parameters involved in the modulation of the saturation state at the site of calcification like DIC concentrating mechanisms and $[Ca]_{CF}$.

Krieger et al. (2023) presented the full-width-half-maximum (FWHM) parameter which has been calibrated in aragonite as a proxy for saturation state (DeCarlo et al. 2017), no quantitative but qualitative analyses can be done when applied
to calcite which is the case here. In our subset of data there was no significant change in FWHM in either of the complexes with increasing irradiances again highlighting a relatively stable saturation state across treatments, in line with $pH_{CF}$ and calcification data.

B/Ca has been used as a proxy for $[CO_3^{2-}]_{CF}$, however this proxy has only been derived for aragonite so no quantitative estimate can be made here but can be used as a potential indication of changes in the carbonate parameters in the calcification
space. No relationship is observed for the geniculate complexes of B/Ca with irradiance. Nevertheless, non-geniculate complexes present significant increase in B/Ca with increasing irradiances (parabolic for *Pneophyllum complex*, positive for *Phymatolithopsis complex*), which could highlight changes in the DIC pool (eg. decreasing $[CO_3^{2-}]_{CF}$ with increasing irradiance). Differences within the non-geniculate complexes are also observed with B/Ca $_{Phymatolithopsis\ complex}$ < B/Ca$_{geniculate}$< B/Ca $_{Pneophyllum\ complex}$ (eg. $[CO_3^{2-}]_{CF\ Phymatolithopsis\ complex}$ > $[CO_3^{2-}]_{CF\ geniculate}$ > $[CO_3^{2-}]_{CF\ Pneophyllum\ complex}$). In a similar way, U/Ca in
mineral is dependent on solution $[CO_3^{2-}]$ (DeCarlo et al., 2015), no significant change is observed with irradiance but significant changes are observed between morphologies, U/Ca$_{geniculate}$ > U/Ca$_{non-geniculate}$ implying different modulation of $[CO_3^{2-}]_{CF}$, $[CO_3^{2-}]_{CF\ geniculate}$ < $[CO_3^{2-}]_{CF\ non-geniculate}$. This overall highlights a lower $DIC_{CF}$ in geniculate relative to the non-geniculate complexes, as for similar $DIC_{CF}$, higher $pH_{CF}$ should increase $[CO_3^{2-}]_{CF}$ but this not observed here.

If $pH_{CF}$ is maintained but $DIC_{CF}$ is modulated then compensatory mechanisms would be needed to sustain a stable
saturation state in those two complexes at the complex level and with changing irradiances, this could be achieved through $[Ca]_{CF}$ modulations.

Mg/Ca is another parameter that could be used to infer the $[Ca]_{CF}$ following the approach of Krieger et al. (2023). The rationale is that the Mg/Ca ratio of the mineral reflects the Mg/Ca ratio of the precipitating fluid, and that only [Ca] modulates this ratio due to its incorporation within the mineral. However, the presence of organics also influences [Ca] and
[Mg], and there are additional controls on Mg incorporation like temperature (Williams et al., 2014) or change in precipitation rate (Gabitov et al., 2014) so a direct translation of Mg/Ca to $[Ca]_{CF}$ can be too simplistic. Nevertheless, a significant effect of irradiance on Mg/Ca is observed in three out of the four complexes. Different Mg/Ca responses can be observed, positive for *Corallina/Arthrocardia fine*, parabolic for *Pneophyllum complex* and threshold positive for *Phymatolithopsis complex*. Those responses are similar to the B/Ca responses for the non-geniculate complexes. This implies that when $[Ca]_{CF}$ decreases (eg.
Mg/Ca increases), $[CO_3^{2-}]_{CF}$ also decreases (B/Ca increases) and that there is no compensation of changes in $[Ca]_{CF}$ by changing $[CO_3^{2-}]_{CF}$. The fact that variations have similar responses can also highlight the changes in $[Ca]_{CF}$ (eg. driving changes in both Mg/Ca and B/Ca ratios).





### 4.6 Differences of calcification space chemistry between geniculate and non-geniculate complexes

It is clear that the two morphologies have characteristic geochemical parameters and physiological responses (PCA and box plots, Figs. 4 and 8). We have shown that non-geniculate complexes have higher calcification (Kreiger et al., 2023), higher $\delta^{13}C_{mineral}$, lower gross photosynthesis and lower $pH_{CF}$ compared to geniculate species. From those results few differences between morphologies can be highlighted, 1) there is a decoupling between net calcification and gross photosynthesis, higher gross photosynthesis in the geniculate complexes does not translate in higher calcification relative to the non-geniculate complexes, 2) $\delta^{13}C_{mineral}$ reflects different DIC source between the two morphologies, $\delta^{13}C_{mineral}$ is not positively correlated with gross photosynthesis when comparing between morphotypes but it is at the complex level across experimental treatments, 3) despite a lack of relationships between $pH_{CF}$ and changing irradiance at the complex level, non-geniculate and geniculate complexes have two different photosynthetic regimes that could correlate with the $pH_{CF}$ observed, higher $pH_{CF}$ is observed along higher gross photosynthesis in geniculate complexes (Figures 7, 8), 4) there is a decoupling between $pH_{CF}$ and net calcification, higher $pH_{CF}$ does not translate to higher net calcification (Figures 7, 8). Net calcification reflects gross calcification and gross dissolution, so it is not abnormal to see net calcification decoupled from physiological or geochemical data. However, from our data it seems that $pH_{CF}$ is not the limiting parameter of calcification.

If the Mg/Ca ratio reflects the $[Ca]_{CF}$, then the higher Mg/Ca ratio observed in the geniculate complexes suggests a lower $[Ca]_{CF}$. Then this lower calcium concentration appears to be compensated by an increase in the $pH_{CF}$ of the calcification fluid (Figure 8). In contrast, the non-geniculate forms show lower Mg/Ca ratios, implying a higher $[Ca]_{CF}$ and, correspondingly, a lower $pH_{CF}$. This could imply a coupling between $[Ca]_{CF}$ and $pH_{CF}$, potentially through proton exchangers like $Ca^{2+}$-ATPase or other Ca concentrating mechanisms.

Building on previous studies on $\delta^{13}C_{tissue}$, we interpret the changes in $\delta^{13}C_{mineral}$ to reflect changes in the source of DIC. We suggest that higher photosynthetic activity (i.e. gross photosynthesis) observed for the geniculate species implies higher need for DIC to support both photosynthesis and calcification. To compensate for the higher $CO_2$ drawdown of photosynthesis and support calcification other sources of DIC like $CO_2$ diffusion or a better recycling of metabolic $CO_2$ may be involved. Those sources would explain the lower $\delta^{13}C_{mineral}$ in geniculate complexes compared to non-geniculate. Higher photosynthetic activity in the geniculate complexes would supply energy to the metabolism, the trade off potentially being DIC limited calcification.

On the other hand, non-geniculate complexes are relying on fast calcification, the lower photosynthesis activity might limit $CO_2$ drawdown which will allow higher inernal DIC availability and sustain higher calcification. The other argument for DIC being the limiting parameter is the non-variation of $pH_{CF}$ with changing irradiance. While higher $pH_{CF}$ can be achieved for the geniculate through higher photosynthesis activity, the $pH_{CF}$ of non-geniculate complexes are also elevated relative to seawater despite lower photosynthesis activity.

Future research will benefit from indirect (e.g. proxies) and direct constraint (e.g. microelectrode) on $DIC_{CF}$ to test those hypotheses. The geochemical differences between morphologies we observed during this study reflect different





photosynthetic strategies and metabolic needs of the organisms. Here we tried to draw some mechanistic explanation to the observed changes in calcification based on the geochemical differences between non-geniculate and geniculate complexes. We show that $DIC_{CF}$ is a limiting parameter to calcification, we hypothesized that geniculate species have greater passive $CO_2$ diffusion/recycling, while DIC is not as limiting for the non-geniculate due to better carbon concentration mechanisms and lower photosynthetic $CO_2$ drawdown which supports higher rates of calcification. The coralline red algae do present a certain plasticity in their carbon sources for DIC (Bergstrom et al., 2020), focusing on how those mechanisms are impacted by environmental stressors such as OA or temperature will be crucial to understand the impact of global changes on those foundational species.

### 4.7 Does light impact proxies for paleoreconstruction?

Rhodoliths are carbonate structures produced by coralline algae that can be used as archives for paleoreconstruction (eg. MacDonald et al., 2024). The main geochemical differences in our study are observed between the different morphologies of coralline red algae. However, rhodoliths are generally produced by non-geniculate (encrusting) species. Taking that into account, we will focus on the non-geniculate complexes for the rest of this section.

As we observed in the other sections, $\delta^{11}B$-derived $pH_{CF}$ is not impacted by light at the complex levels which does not produce additional complexity the use of the proxy. Anagnostou et al. (2019) presented a robust calibration of the $\delta^{11}B$ proxy based on culture experiments on a high-latitude crustose coralline red algae *Clathromorphum compactum*. As rhodoliths usually are produced by a mix of species, a complex-specific response to ocean acidification and the strong control they exert on their calcification fluid could be a limitation of the proxy, but our findings suggest $\delta^{11}B$ should be at least insensitive to light levels.

Despite significant relationships for Mg/Ca (*Pneophyllum complex* and *Phymatolithopsis complex*) and Li/Ca (*Pneophyllum complex*), Li/Mg ratios did not show any significant effect of changing irradiance, which does not impair the applicability of the temperature proxy for both species. Also, no significant differences were observed for the Li/Ca of the two non-geniculate species.

Our results on low-light adapted species show that light does not impair the application of the $\delta^{11}B$ and Li/Mg proxies.

### 5 Conclusions

The geochemistry ($\delta^{11}B$, $\delta^{13}C_{mineral}$ and trace elements) of four low-light adapted complexes of coralline red algae cultured under different irradiances was investigated in this study following prior work by Krieger et al. (2023). Two morphologies were investigated: geniculate (branching) complexes, *Corallina/Arthrocardia robust* and *Corallina/Arthrocardia fine* and non-geniculate (encrusting/mounding) complexes, *Pneophyllum complex* and *Phymatolithopsis complex*.



The first purpose of this study was to investigate the effect of light (changing irradiance) on the pH of calcification for the different complexes. Based on photophysiological parameters (i.e. gross photosynthesis, ETR max) and $\delta^{13}C_{mineral}$, we show that at the complex levels photosynthesis activity has an impact on the geochemical signature of the mineral. However, despite increasing photosynthetic activity with irradiance, $\delta^{11}B$ or $pH_{CF}$ was maintained constant for all treatments. $pH_{CF}$ was

upregulated relative to seawater in all complexes with complex-specific $pH_{CF}$. No significant effect of light was observed at the complex level (in the range of irradiance 0.6-2.3 mol photons $m^{-2} d^{-1}$).

The main differences in physiological and geochemical parameters are observed between morphologies. Those results demonstrate two calcification regimes. We show that non-geniculate complexes have higher net calcification, higher $\delta^{13}C_{mineral}$, lower gross photosynthesis, lower $pH_{CF}$, lower Mg/Ca while geniculate have lower net calcification, lower $\delta^{13}C_{mineral}$, higher

gross photosynthesis, higher $pH_{CF}$, higher Mg/Ca.

We highlight that $pH_{CF}$ can be positively influenced via photosynthetic regimes (inherent to morphologies). We show that net calcification is decoupled from $pH_{CF}$ and that based on Mg/Ca, changes in $pH_{CF}$ are compensated by changes in $[Ca]_{CF}$. The main differences between calcification modes is likely due to DIC and carbon concentrating mechanisms reflected in our data by $\delta^{13}C_{mineral}$. The lower $\delta^{13}C_{mineral}$ of geniculate species can indicate a relatively more important contribution of passive

$CO_2$ diffusion and/or higher recycling of $CO_2$ to the DIC pool.

Higher calcification in non-geniculate complexes is supported by higher $DIC_{CF}$ due to lower $CO_2$ drawdown from photosynthesis and efficient carbon-concentrating mechanisms. Additionally, despite lower photosynthetic activity compared to geniculate complexes, photosynthesis-independent processes may help maintain elevated $pH_{CF}$ reducing the energetic cost of pH regulation. In contrast, geniculate complexes experience greater $CO_2$ drawdown limiting $DIC_{CF}$ use for calcification.

Although $CO_2$ recycling or passive diffusion may partly offset this limitation, the energy obtained from photosynthesis in geniculate complexes is likely prioritized to other metabolic needs at the expense of calcification. These differences could be explained by the competition experienced by non-geniculate species to not be overgrown (e.g. turf algae) which must also rely on fast calcification while geniculate species must compensate for a more dynamic environment and prioritize other needs (e.g. grazing, repairs) (Stenneck et al., 1986; Connell, 2003b; Edwards and Connell, 2012).

No effect of irradiance is observed on the temperature proxy Li/Mg for the different complexes in the range of irradiances tested in this study. Light should not add additional complexity to the interpretation of the Li/Mg and $\delta^{11}B$ proxies when applied to paleoreconstruction studies from rhodolith beds.

Development of proxies to derive a second carbonate parameter in high Mg calcite such as the $[CO_3^{2-}]_{CF}$ proxies (e.g. B/Ca, U/Ca) developed in the aragonitic corals as well as direct microelectrode measurements of the calcifying parameters

(e.g. $pH_{CF}$, $DIC_{CF}$) will be relevant to study the dynamics of the calcification space in coralline red algae.

This study complements the work of Krieger et al. (2023), building upon their findings to further explore the calcification physiology of coralline algae and the role of light in these processes. This study demonstrates variability in responses of coralline red algae under irradiance and highlights distinct biomineralization mechanisms between branching



(geniculate) and encrusting (non-geniculate) low-light adapted complexes. Photosynthesis impacts the availability and source
of $DIC_{CF}$ which has implications on calcification. In the perspective of calcification, plasticity on DIC sources is determinant
for acclimation of coralline red algae. Additional study on the joint effect of OA and changing irradiance might provide some
interesting dynamics and will be needed to understand the full implications of future global changes and associated
perturbations on the coralline algae communities and dependent ecosystems.

**Acknowledgement:** We thank Seth John, Josh West, Shun-Chun Yang for technical support and use of the Neptune at the
Dornsife Plasma facility at University of Southern California. We thank Céline Liorzou, Marie-Laure Rouget, Bleuenn
Guéguen, Oanez Lebeau, Fabien Dewilde for technical support and use of the instruments at the Pôle Spectrométrie Océan at
the Institut Européen de la Mer (Plouzané, France).

**Funding sources:** This work was funded by a grant from the David and Lucile Packard Foundation (grant no. 85180), National
Science Foundation grant no. NSF-RISE-2024426, and by gifts from Ocean kind and Dalio Philanthropies. The Center for
Diverse Leadership in Science is also supported by grant no. NSF-RISE-2228198, the Waverley Street Foundation, and the
Sloan Foundation.

**Author contributions:** C.C. and R.A.E. conceived the project. C.C and R.A.E directed the research. E. K. and C.C. performed
culturing experiments, specimen characterization. M.G. performed isotope and trace element analyses at USC and IUEM. M.G
and J.G. performed statistical analyses and figures. M.G., R.A.E., C.C., and E.K. interpreted the geochemical data. M.G. wrote
the manuscript with input from R.A.E, C.C and E.K. All authors read and edited the manuscript.

**Conflicts of Interest:** The authors declare no conflict of interest. The funders had no role in the design of the study; in the
collection, analyses, or interpretation of data; in the writing of the manuscript, or in the decision to publish the results.

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

Gulf Profe








## Figure Caption

**Figure 1:** Pictures of the four coralline red algae complexes used in this study (already presented in Krieger et al., 2023) and
showing the different morphologies: non-geniculate (e.g. crustose) and geniculate (e.g. branching). Geniculate complexes: *Corallina/Arthrocardia* "robust" and *Corallina/Arthrocardia* "fine", non-geniculate complexes: *Pneophyllum* complex and *Phymatolithopsis* complex.

**Figure 2:** Averages of photophysiological parameters of the four complexes from Krieger et al. (2023) against irradiances.
A. Net calcification ($mg_{CaCO3}/cm^2/day$), B. Gross photosynthesis ($\mu gO_2/cm/h$), C. Maximum electron transport rate, ETRmax, D. Photosynthetic efficiency measured by the "variable fluorescence" normalized to maximum fluorescence, Fv/Fm, E. Chlorophyll a, Chl a (mg/g). Averages are calculated from the full dataset from Krieger et al. (2023), error bars are based on 2 SD. Regressions are shown in Figure S2.

**Figure 3:** Averages of geochemical data measured in this study against irradiances. A. Net calcification ($mg_{CaCO3}/cm^2/day$), B. boron isotopes of the mineral, $\delta^{11}B$ (‰), C. carbon isotopes of the mineral $\delta^{13}C_{mineral}$ (‰), D. carbon isotopes of the tissue $\delta^{13}C_{tissue}$ (‰) from Krieger et al. (2023), E. B/Ca of the mineral ($\mu mol/mol$) and F. Mg/Ca of the mineral mmol/mol). Error bars are based on 2 SD. Regressions are shown in Figure S1.

**Figure 4:** Mantel test and principal component analysis (PCA) of the geochemical and photo physiological data used in this study.

**Figure 5:** Multi-panel plots showing crossplots of $\delta^{13}C_{mineral}$ (‰) and $\delta^{11}B$ (‰). Averages are calculated based on this study for geochemical parameters and from the full dataset in Krieger et al. (2023). Individual paired data are also shown to
maximize the information displayed, color scheme corresponds to the different irradiances. A. crossplot of $\delta^{13}C_{mineral}$ (‰) and $\delta^{13}C_{tissue}$ (‰), linear significant relationships are shown with black lines, B. $\delta^{11}B$ (‰) and $\delta^{13}C_{mineral}$ (‰), C. $\delta^{13}C_{mineral}$ (‰) and Net Calcification ($mg_{CaCO3}/cm^2/day$), D. $\delta^{11}B$ (‰) and Net Calcification ($mg_{CaCO3}/cm^2/day$), E. $\delta^{13}C_{mineral}$ (‰) and gross photosynthesis ($\mu gO_2/cm/h$) and F. $\delta^{11}B$ (‰) and gross photosynthesis ($\mu gO_2/cm/h$).

**Figure 6:** $pH_{CF}$ calculated from $\delta^{11}B$ against irradiance for the four complexes, A. *Corallina/Arthrocardia* robust, B. *Corallina/Arthrocardia* fine, C. *Pneophyllum* complex, D. *Phymatolithopsis* complex.

**Figure 7:** Multi-panel plots showing crossplots of $pH_{CF}$, A. net calcification ($mg_{CaCO3}/cm^2/day$), B. gross photosynthesis ($\mu gO_2/cm/h$), C. residual full-width-half-maximum, FWHM, D. $\delta^{13}C_{mineral}$ (‰) and E. Mg/Ca (mmol/mol).



**Figure 8:** Box plots comparing geniculate complexes (blue) and non-geniculate (green). Box plots show the median,10, 90 percentiles and as well as the individual data points.



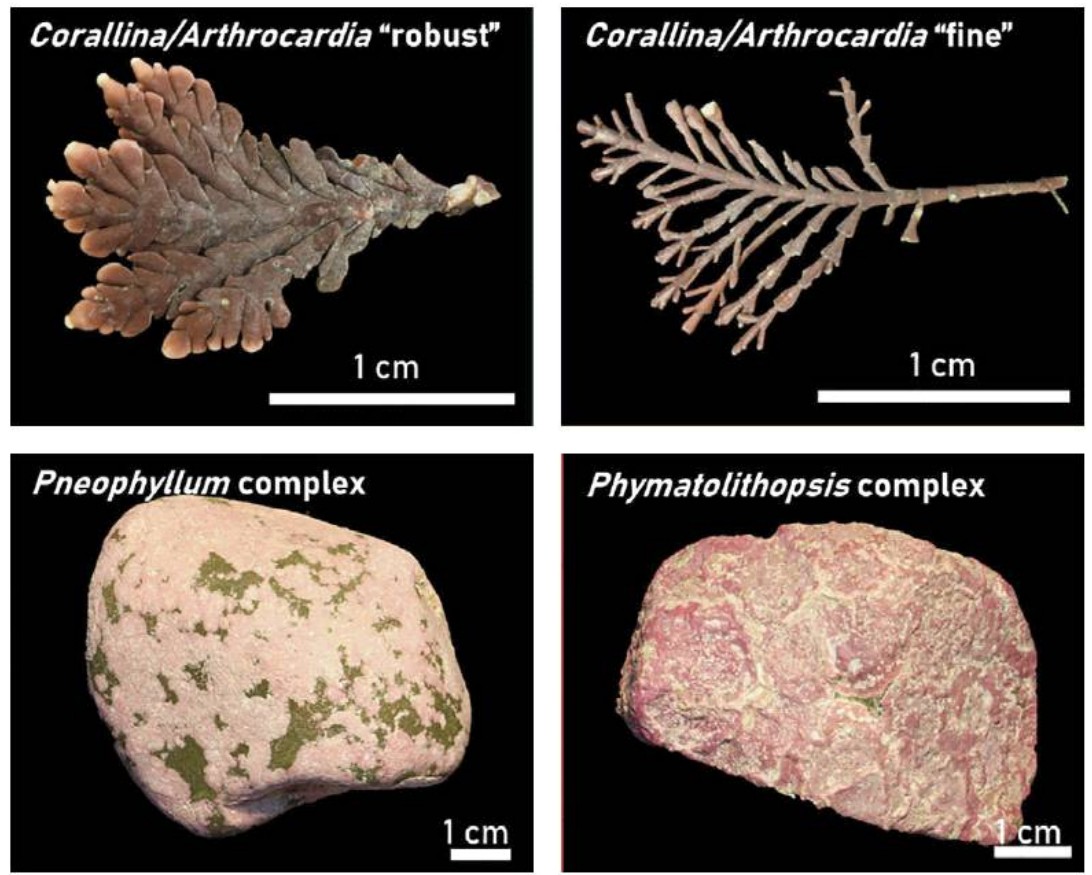

f01

**Figure 1:** Pictures of the four coralline red algae complexes used in this study (already presented in Krieger et al., 2023) and showing the different morphologies: non-geniculate (e.g. crustose) and geniculate (e.g. branching). Geniculate complexes: *Corallina/Arthrocardia* "robust" and *Corallina/Arthrocardia* "fine", non-geniculate complexes: *Pneophyllum* complex and *Phymatolithopsis* complex.



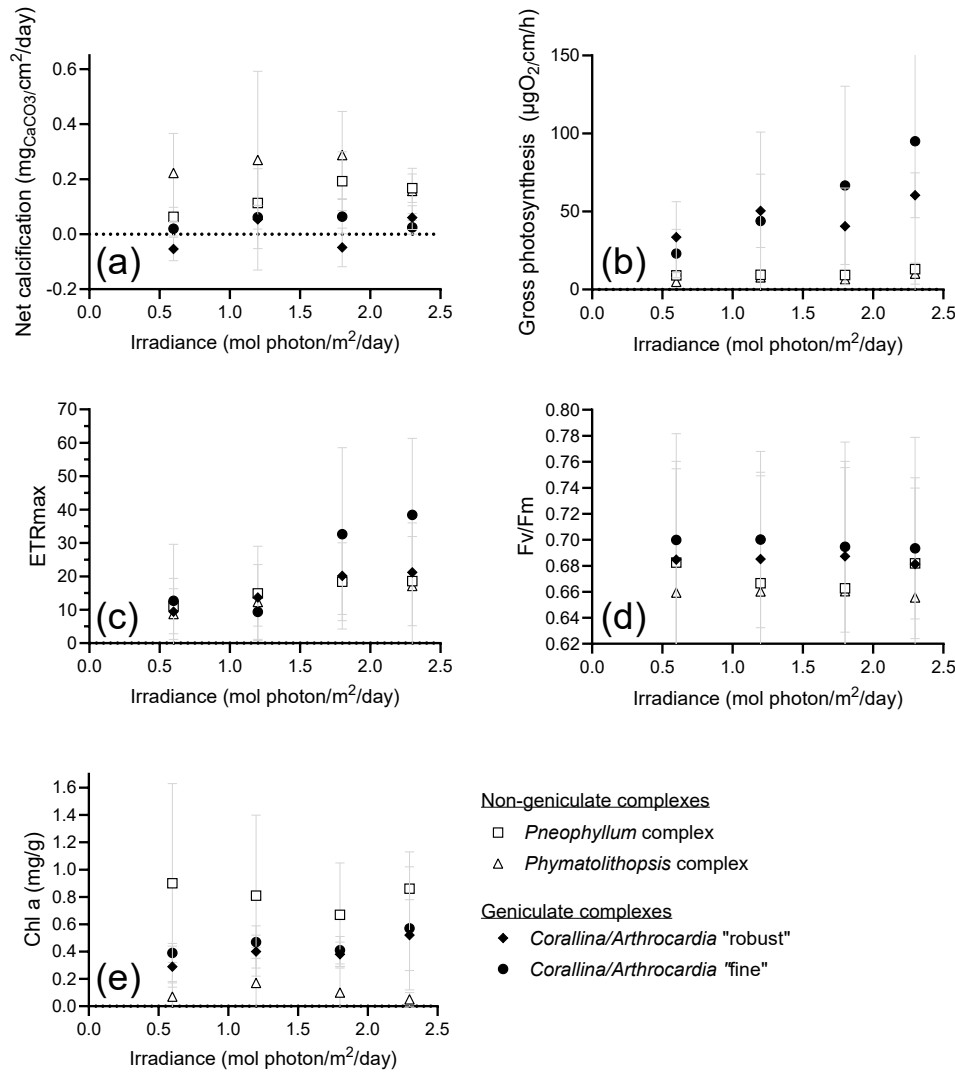

**f02**

**Figure 2:** Averages of photophysiological parameters of the four complexes from Krieger et al. (2023) against irradiances. A. Net calcification ($mg_{CaCO_3}/cm^2/day$), B. Gross photosynthesis ($\mu gO_2/cm/h$), C. Maximum electron transport rate, ETRmax, D. Photosynthetic efficiency measured by the "variable fluorescence" normalized to maximum fluorescence, Fv/Fm, E. Chlorophyll a, Chl a (mg/g). Averages are calculated from the full dataset from Krieger et al. (2023), error bars are based on 2 SD. Regressions are shown in Figure S2.







f03

**Figure 3:** Averages of geochemical data measured in this study against irradiances. A. Net calcification ($mg_{CaCO3}/cm^2/day$), B. boron isotopes of the mineral, $\delta^{11}B$ (‰), C. carbon isotopes of the mineral $\delta^{13}C_{mineral}$ (‰), D. carbon isotopes of the tissue $\delta^{13}C_{tissue}$ (‰) from Krieger et al. (2023), E. B/Ca of the mineral (μmol/mol) and F. Mg/Ca of the mineral mmol/mol). Error bars are based on 2 SD. Regressions are shown in Figure S1.





**Figure 4:** Mantel test and principal component analysis (PCA) of the geochemical and photo physiological data used in this study.





**Figure 5:** Multi-panel plots showing crossplots of $\delta^{13}C_{mineral}$ (‰) and $\delta^{11}B$ (‰). Averages are calculated based on this study for geochemical parameters and from the full dataset in Krieger et al. (2023). Individual paired data are also shown to maximize the information displayed, color scheme corresponds to the different irradiances. A. crossplot of $\delta^{13}C_{mineral}$ (‰) and $\delta^{13}C_{tissue}$ (‰), linear significant relationships are shown with black lines, B. $\delta^{11}B$ (‰) and $\delta^{13}C_{mineral}$ (‰), C. $\delta^{13}C_{mineral}$ (‰) and Net Calcification (mg$_{CaCO3}$/cm$^2$/day), D. $\delta^{11}B$ (‰) and Net Calcification (mg$_{CaCO3}$/cm$^2$/day), E. $\delta^{13}C_{mineral}$ (‰) and gross photosynthesis (μgO$_2$/cm/h) and F. $\delta^{11}B$ (‰) and gross photosynthesis (μgO$_2$/cm/h).

f05



**f06**

**Figure 6:** pH$_{CF}$ calculated from $\delta^{11}$B against irradiance for the four complexes, A. *Corallina/Arthrocardia* robust, B. *Corallina/Arthrocardia* fine, C. *Pneophyllum* complex, D. *Phymatolithopsis* complex.





**Figure 7:** Multi-panel plots showing crossplots of pH$_{CF}$, A. net calcification (mg$_{CaCO3}$/cm$^2$/day), B. gross photosynthesis (μgO$_2$/cm/h), C. residual full-width-half-maximum, FWHM, D. δ$^{13}$C$_{mineral}$ (‰) and E. Mg/Ca (mmol/mol).



**Figure 8:** Box plots comparing geniculate complexes (blue) and non-geniculate (green). Box plots show the median,10, 90 percentiles and as well as the individual data points.