# Peer review of "The influence of irradiance and interspecific differences on $\delta^{11}$ B, $\delta^{13}$ C"

_EGUsphere, 2025_

## Author Comment (AC1)

[revised manuscript text omitted]
 (mgCaCO3/cm²/day), D. δ11B (‰) and Net Calcification (mgCaCO3/cm²/day), E. δ13Cmineral (‰)
- 745 and gross photosynthesis ( $\mu g O_2/cm/h$ ) and F.  $\delta^{11}B$  (‰) and gross photosynthesis ( $\mu g O_2/cm/h$ ).

746

- Figure 6: pHCF calculated from  $\delta^{11}$ B against irradiance for the four complexes, A. Corallina/Arthrocardia robust, B.
- 748 Corallina/Arthrocardia fine, C. Pneophyllum complex, D. Phymatolithopsis complex. Average values per treatment are
- presented with 2 SD error bars. Individual datapoints are also presented to assess variability within treatment.

750

- 751 **Figure 7:** Multi-panel plots showing crossplots of pHCF, A. net calcification (mgCaCO3/cm2/day), B. gross photosynthesis
- 752 (μgO2/cm/h), C. residual full-width-half-maximum, FWHM, D. δ13Cmineral (‰) and E. Mg/Ca (mmol/mol). Large symbols
- show averages derived from full dataset from Krieger et al. (2023) while small colored symbols show individual paired data
- and irradiance level to display maximum information. Error bars are shown as 2 SD.

[revised manuscript text omitted]

**Supplemental Figures**

Figure S1: Data and significant models (black line) for the geochemical parameters measured and used in this study.

Figure S2: Data and significant models (black line) for the physiological parameters from Krieger et al. (2023) and used in this study.

**Figure S3:** Principal component analyses for (a) the relevant geochemical and physiological parameters used in this study and (b) elemental ratios and physiological parameters.

**Figure S4:** (a-d) Correlation matrices providing pairwise correlations between geochemical, and photo physiological data for a. *Corallina/Arthrocardia* "robust", b. Corallina/Arthrocardia "fine", c. *Pneophyllum* complex and d. *Phymatolithopsis* complex.

Figure S5: Correlation matrices for (a) the geniculate complexes and (b) the non-geniculate complexes.

**Figure S6:** Cross-plots of  $\delta^{13}$ Cmineral and  $\delta^{11}$ B for other photo-physiological parameters, (a) and (b) Gross photosynthesis, (c) and (d) for ETRmax, (e) and (f) for Chl a.

**Figure S7:** Cross-plots of B/Ca with (a)  $\delta^{11}$ B and (b) Chl a.

**Supplemental Tables**

Table S1: Geochemical and physiological data.

Table S2: Comparison of linear and quadratic models based on AIC for the geochemical parameters measured in this study.

**Table S3:** Comparison of linear and quadratic models based on AIC for the physiological parameters published in Krieger et al., (2023).

**Table S4:** ANOVA testing geochemical and physiological data against changing irradiance.

**Table S5:** T-test for parameters presenting significant ANOVA with changing irradiance (from Table S4).

**Table S6:** ANOVA testing geochemical and physiological data between complexes.

**Table S7:** T-test for parameters presenting significant ANOVA when testing for differences between complexes.

**Table S8:**  $\delta^{11}$ B of NIST 8301, JCp-1 and seawater measured in this study.

**Physiological Data PCA**

**Trace Element Data PCA**

fS3

**Correlation matrices**

**Correlation matrices**

**Geniculate complexes**

**Non-geniculate complexes**

**Non-geniculate complexes**

☐ Pneophyllum complex

Phymatolithopsis complex

**Geniculate complexes**

- Corallina/Arthrocardia "robust"
- Corallina/Arthrocardia "fine"
- Irradiance 0.6
- Irradiance 1.2
- Irradiance 1.8
- Irradiance 2.3

---

## Author Comment (AC3)

Dear editor and reviewers,

We would like to thank you for your detailed and constructive comments that helped to improve the manuscript. We provide detailed point by point responses to your comments below, as well as version of our revised manuscript with changes made highlighted in yellow for reviewer 1 and in green for reviewer 2.

Best Regards,

Maxence Guillermic on behalf of all co-authors.

**Reviewer 1**

Dear Dr. de Winter,

I was pleased to review manuscript # 2025-2626 "The influence of irradiance and interspecific differences on  $\delta$ 11B,  $\delta$ 13C and elemental ratios in four coralline algae complexes" by Guillermic and colleagues. The manuscript represents a significant contribution to the understandings of coralline algal geochemistry and calcification mechanisms, which are still poorly understood, especially given the growing number of identified species and the small research group that studies them. I agree with the authors that this study represents part of the groundwork required to validate the use of certain paleoenvironmental proxies. The major findings of insignificant effects of irradiance on δ11B, δ13C and elemental ratios in four coralline algae species, but also notable differences in DIC modulations between geniculate and non-geniculate species represents an important step towards understanding calcification mechanisms and biological processes among diverse coralline algal morphologies and species. The manuscript is well presented, clear, and data support the findings. Except for a few technical corrections, figures clearly demonstrate findings and support interpretations. It is rare that a paper includes this abundance of data collected and from multiple species. I recommend that the manuscript be accepted subject to minor corrections.

I remain available if you have any questions.

Regards

\_\_

Scientific signifiance: Excellent

Scientific quality: Excellent

Presentation quality: Good

Reviewer Recommendation: Accepted subject to minor revisions

\* I would not be willing to review the revised manuscript.

\_\_

**General Comments:**

 I think the manuscript would benefit with a higher impact statement in the introduction and abstract on what the potentially risks are of not understanding irradiance impacts on calcification and geochemistry (e.g., erroneous paleoenvironmental reconstructions, linking the wrong parameter to calcification rates, providing conflicting data between different archives and creating doubt in paleo-environmental timeseries, and potentially even more importantly, environmental forecasts which allow us to put in place the proper environmental policies and protections, etc.). This would better elucidate the importance of the study.

**Response:** We added in the abstract "In this context, evaluating the effect of oceanic change and photo-physiological parameters on geochemical proxies is critical, as such gaps may lead to erroneous paleoenvironmental reconstructions, misattributed drivers of calcification responses, and ultimately compromise conservation strategies." And in the introduction: "To increase the reliability of coralline algae for paleoclimate reconstruction " and "This is critical as erroneous interpretation of proxies can undermine confidence in long-term environmental records, drivers of calcification and compromise forecasts that inform marine policy and conservation strategies."

**Specific Comments:**

- The point made on line 456 is very interesting: "These differences could be explained by the competition experienced by non-geniculate species to not be overgrown (e.g. turf algae) which must also rely on fast calcification while geniculate species must compensate for a more dynamic environment and prioritize other needs (e.g. grazing, repairs)".
  - A short description on morphologies and "behaviour" (e.g., rolling / mobile vs. encrusting / immobile) could be highlighted somewhere in the introduction to "foreshadow" this point.

**Response:** We added at the beginning of the introduction: "Coralline red algae show two main morpho-functional groups, geniculate and non-geniculate. Geniculate corallines have non calcified joints that connect the calcified intergenicula to allow for higher thallus flexibility, distinct morphological traits allow them to grow in various habitats and to cope with a wide range of environments (Noisette et al., 2013; McCoy and Kamenos, 2015).".

• In the methods section 2.3., I was unclear about the temporal span represented by these algae samples that were powdered. Were spores deposited in the water column to cultivate the algae from scratch or were small crusts/branches collected and placed in tanks experiments? If the latter, provide growth rates of species and the time represented by each species / samples. Clarifying which growth layers are powdered for analysis would also be important to know (specific language on whole samples including epithallus and perithallus). Could different temporalities represented by the different species be responsible for some of the geochemical differences reported here?

**Response:** After collection, samples were stained with alizarin red and only material above the stain line was sampled. Additionally, we sampled material (e.g., growth margins from crustose corallines) that we knew grew in the lab as they were growing on the epoxy that we used initially to form bases for the articulates and cover up "unwanted" crusts growing on the pebbles together with the target species. The whole thallus was sampled and no specific tissue layer was targeted. We added line 132: "Samples were stained with alizarin red and only material above the stain line was sampled to ensure sampling the new growth."

From Krieger et al. (2023): "For collection, geniculate corallines were chiseled from the rock, retaining the attached crust to avoid damaging them. Cobbles or rocks covered with thick or smooth crusts were collected directly from the seafloor. [...] After collection, organisms were transported to the laboratory facilities within 20 min in cooler bins filled with ice and cool packs to further minimize thermal and light stress. At the laboratory, organisms were kept under low light levels (daily dose 0.06–0.23 mol photons m\_2 d\_1) for 2 d to allow for slow acclimation to laboratory conditions. Subsequently, organisms were carefully physically cleaned of epibionts and labeled according to the morpho-anatomic classification. Epoxy (Z-Spar A-788 Splash Zone) was used to form a base for geniculate coralline algae and to cover crusts of other species on cobbles/rhodoliths. Specimens were then distributed into the experimental tanks."

 The introduction would benefit from a short theoretical explanation on boron, carbon, Li and Mg fractionation and how it is affected by seawater and calcification etc. Some of this information is found in the discussion and methods, but should be discussed in the introduction.

**Response:** For the purpose of this study, we chose to keep the introduction concise and focused on the broader context and objectives as further information can be found in the Methods and Discussion. However, we added in the introduction one sentence: "Boron isotopes have been developed in carbonate as a proxy of pH in the fluid that it is precipitated within. The sensitivity of the  $\delta 11B$  proxy to pH is based on the predominant incorporation of borate ion in the carbonate structure (Hemming and Hanson, 1992).

Line 114: Ref (Gaillardet et al., 2001, Wang et al., 2008). Indicate if the more specific procedure is found in these papers. If not, add the time(s) of dissolution etc.

**Response:** References from this paragraph give all the information for sample preparation.

• In the introduction and discussion (maybe section 4.6), there is a missing statement on why this specific study was conducted. Something like, "In other words, previous studies such as Anagnostou et al. found [...] but lacked understandings about [...]. This understanding is critical because without it we risk [...]." Or focus on Comeau et al. 2019's findings and state something like "To further test Comeau et al's

hypothesis we investigated calcification differences between faster and slower growing coralline algae species using geochemical tracers.

 We need something clear that states how this paper builds on what is already known.

**Response:** We took your suggestion and added at the end of the introduction: "To further test Krieger et al's and Comeau et al's hypothesis we investigated calcification differences between faster and slower growing coralline algae complexes using geochemical tracers".

o In the discussion: a comment can be added about how internal pH has been studied here and additional studies on how coralline algae modulate pH CF, DIC CF calcification would be helpful to capture the limits of plasticity of photosynthesis and calcification modulation with increasing ocean acidification as to provide limits or warnings for policy application?

**Response:** We added at the end of section 4.6: "The coralline red algae do present a certain plasticity in their carbon sources for DIC (Bergstrom et al., 2020) and regulation of pHCF, which can provide some resilience to changing environmental conditions. Additional studies on how coralline algae modulate DICCF and pHCF would be helpful to capture the limits of plasticity of photosynthesis and calcification modulation under stressors such as ocean acidification or warming temperature. This understanding will be critical for assessing the impact of global changes on those foundational species."

 Section 4.7: could you add that this study also supports that well defined DNA work might be required to calibrate geochemical data to the species for paleoenvironmental reconstructions?

**Response:** Thank you for this suggestion. We added: "Nevertheless, with the increasing availability in species-specific geochemical data, a rigorous approach may involve using DNA-based identification within the core to calibrate geochemical records."

• Figure 4: Provide a more elaborate description especially for panels A-D. Consider also adding a label to colour bar (axis)

**Response:** We made a mistake here, we are presenting a correlation matrix not a Mantel test. Mantel tests compare two distance matrices and output Mantel's R, which is a bit different from pairwise correlations between variables. Those correlation matrices are now presented in the Supplemental information.

• Figure 7: Explanation for larger symbols and error bars.

**Response:** Figure caption now reads: "Multi-panel plots showing crossplots of pHCF, A. net calcification (mgCaCO3/cm2/day), B. gross photosynthesis ( $\mu$ gO2/cm/h), C. residual full-width-half-maximum, FWHM, D.  $\delta^{13}$ Cmineral (‰) and E. Mg/Ca (mmol/mol). Large symbols show averages derived from full dataset from Krieger et al. (2023) while small colored symbols

show individual paired data and irradiance level to display maximum information. Error bars are shown as 2 SD.".

**Technical Corrections:**

• Line 70: "CCA" This is the only place where this acronym is used. It is also not explained anywhere

**Response:** We changed for "low-light coralline algae complexes" and do not use CCA within the manuscript.

• Line 162: "However, Krieger et al. (2023) presented two significant relationships, one non-linear for Corallina and one non-linear for Spongites when the full dataset was taken into account". ◊ Clarify if this refers to a relationship between net calcification and irradiance. If so, consider inversing the order of the two last sentences of the paragraph.

Response: We changed the order.

• Line 180 to 181, when possible, always add geniculate or non-geniculate adjectives before species to orient readers. You do this most of the time, but check throughout

**Response:** noted, we added when clarification was needed.

• The two back-to-back sentences "There are no significant linear relationships between  $\delta 11B$  and irradiance (Tables S3 and S4). No significant linear or non-linear regression was observed between  $\delta 11B$  and irradiance" are a bit redundant

**Response:** This has been reduced to "No significant linear or non-linear regression was observed between  $\delta^{11}$ B and irradiance (Tables S3 and S4)".

• Sometimes e.g., is used instead of the correct i.e., (e.g., lines 369 and 371.

**Response:** We changed for "i.e." when needed throughout the text.

• Line 412: make sure to define OA (ocean acidification). I think this is the only place the acronym is used

**Response:** We removed the acronym throughout the text.

- Figure 3: correct the alignment of B)
  - I think the font of the axes and axes titles might also be smaller

**Response:** We corrected the alignment of panel (b), the font 13 is however the same as the other figures, but this is due to the distortion to fit Biogeosciences standards over the submission, we will make sure that figures have the same format.

Reviewer 2

Guillermic et al. (2025) seek to address a crucial literature gap by studying experimental evidence of the effect of varying levels of irradiance on growth in geniculate versus nongeniculate species complexes of algae. They use data from tank studies conducted by Krieger et al. (2023) where two geniculate complexes and two non-geniculate complexes of coralline algae were collected from two sites in Te Moana-o-Raukawa Cook Strait, Te Whanganui a Tara Wellington, Aotearoa New Zealand. All complexes were subjected to four different irradiance treatments representing naturally occurring levels at the site (0.6, 1.2, 1.8 and 2.3 mol photons/m²/day) with corresponding fluctuations in irradiance to account for the diurnal insolation pattern. Various isotope measurements and elemental ratios were calculated including d¹³C, d¹8O, d¹¹B, Mg/Ca, Li/Ca, Sr/Ca, U/Ca and Li/Mg. Additionally, gross photosynthesis along with other parameters of photosynthetic efficiency, net calcification and d¹¹B-derived pHCF were measured and/or calculated.

All complexes, except the non-geniculate Phymatolithopsis showed a significant positive correlation between irradiance and parameters measuring photosynthetic activity. Significant positive correlations were also observed between d13Cmineral and irradiance, except in one geniculate complex, Corallina/Arthrocardia robust whose d13Cmineral remained stable across all treatments. d11B-derived pHCF generally stayed constant across all treatments. No significant differences in net calcification were observed across irradiance treatments. However, the most pertinent results contributing to addressing gaps in literature were the different calcification regimes observed between morphologies, where non-geniculate complexes showed higher net calcification and lower pHCF than geniculate complexes despite the latter having higher gross photosynthesis. Further, the authors speculate that the differences in d13Cmineral based on morphology may be due to differences in DIC pools available to the geniculate versus non-geniculate morphologies and possibly individual complexes or differential uptake of CO2 passively or through internal recycling as supported by B/Ca and U/Ca results. Overall, the authors have collected and analyzed a robust set of isotope and elemental data in addition to calculating various parameters for photosynthetic efficiency and calcification to effectively support their conclusions. The sample size and methods used are appropriate for the analyses being conducted and interpretations being made. I would suggest minor edits and request clarification to the text prior to publication to contextualize the study within broader algal literature and ensure balanced communication of the study's results.

**Title**: The title is accurate, concise and descriptive.

• Lines 1 & 2: I would either add that specimens are from a mid-latitude location and temperate climate or indicate the study site/country (i.e., Aotearoa New Zealand)

**Response:** The title now reads: "The influence of irradiance and interspecific differences on  $\delta^{11}$ B,  $\delta^{13}$ C and elemental ratios in four coralline algae complexes from Aotearoa, New Zealand

**Abstract:** Abstract offers an effective and concise summary of the results and discussion.

• Line 13: Change the Arabic numeral "4" to the written "four" following writing conventions (i.e., numbers below nine that are not statistics are spelled out whereas numbers >10 are written as numerals).

**Response:** This is changed. Thank you for the explanation.

**Introduction:**

General comments: The introduction is generally well-written and offers a logical progression discussing the ecosystem-level importance of coralline algae then further expanding on the proxy measurements of pHCF, followed by discussion on the regulation of biogenic calcification by pH, light and photosynthesis. A main point for revision in the introduction would be to adjust the paragraph at lines 48-55 to summarize and reflect more on coralline algae literature and associated knowledge gaps, following which the summary of coral literature could be brought forward (i.e., indicating that the current gap in literature being addressed on the impact of light on coralline algal calcite formation has been explored more extensively in coral literature).

**Response:** We added at the end of this paragraph: "Limited research has been carried out on coralline algae, and although irradiance can impact pHCF of coralline algae (Comeau et al., 2019), much more research is required."

**Specific Recommendations:**

Lines 37 & 38: Awkward phrasing here with the use of the word "some" twice. I
would remove the phrase "but with some evidence" and simply write "evidence
suggests".

**Response:** We changed it.**

• Line 41: Coralline algae have already been used for paleoclimate reconstruction. It may be more accurate to write something to the effect of "To increase the reliability of coralline algae paleoclimate reconstructions, a good understanding..."

**Response: We changed it.**

• Line 43: As pHCF refers to pH of calcifying fluid, explicitly define the acronym as "pH of calcifying fluid (pHCF)" in accordance with writing conventions.

**Response: We changed it.**

Lines 46-47: More recent reference to include would be Cornwall et al. (2020): A
coralline alga gains tolerance to ocean acidification over multiple generations of
exposure.

Response: We added it thank you.

Lines 48-55: This is a good summary of existing coral research, however we are
missing the direct connection to coralline algae in this paragraph. I would suggest a
more concise explanation of coral research in favour of additional background on pH
geochemical tracers in coralline algae including studies cited in the introduction
already (e.g., Donald et al., 2017) or linking existing coral research to emerging
coralline algae research.

**Response:** We added at the end of the paragraph to make the link with coralline algae: "Limited research has been carried out on coralline algae, and although irradiance can impact pHCF of coralline algae (Comeau et al., 2019), much more research is required."

- Lines 56-62: I would also add more explicitly here that there may be differences in light adaptation and calcification mechanisms for species in tropical vs temperate vs polar environments (i.e., latitude and climate play a role) explaining some of this variability, possibly before the sentence on line 58. Gould et al. (2022) and Williams et al. (2018) are some examples for Arctic studies examining the relationship between light and calcification that have not been cited and show that calcification is reduced during periods of low irradiance but still occurs at decreased rates.
  - Williams et al., 2018: Effects of light and temperature on Mg uptake, growth, and calcification in the proxy climate archive Clathromorphum compactum;
  - Gould et al., 2022: Growth as a function of sea ice cover, light and temperature in the arctic/subarctic coralline C. compactum: A year-long in situ experiment in the high arctic).

**Response:** Thank you for suggesting those references. We added at the end of this section: "In contrast, coralline algae in polar regions can continue calcifying at reduced rates even under prolonged low-light conditions associated with seasonal cycles or sea ice cover (Williams et al., 2018; Gould et al., 2022). These latitudinal (e.g. tropical, temperate or polar environments) and climate-driven differences in light adaptation and calcification mechanisms can contribute to the variability reported across studies. Although light clearly affects calcification, the mechanistic links between irradiance, photophysiology, and calcification is not fully understood."

• Line 70: Ensure the acronym, "CCA" is defined prior to using it. There do not appear to be other instances where "CCA" is used in the text, therefore the term can be written in its full form.

**Response:** Acronym was removed throughout the text.

• Line 75:  $\delta^{18}$ O is not discussed in the body of the paper, so should either be excluded here or if there are any relevant results they may be briefly discussed.

Response: We removed it.

**Methods:**

General comments: The methods section is clear and concise. Appropriate methods are used to address the defined research questions with redundancies built into the methodology for robust interpretation. Most concerns here are related to clarity of writing. I would recommend minor changes listed below to follow formal/academic writing conventions. The only concern of significance here is the inclusion of the Mantel test methodology which does not seem to be well explained. While the figure itself offers a useful summary, there is little reference to it in the text. I would recommend elaborating on its purpose in the context of interpretation of results or see further recommendations in the comments on the results section.

**Specific recommendations:**

• Line 84: "Latter" is used incorrectly here. This typically refers to the second of two items in a list (i.e., latter vs former) or the last item in a list. The phrase prior to the comma can be removed and Krieger et al. (2023) can be cited at the end of the sentence in parentheses.

**Response:** We changed the sentence as suggested.**

Lines 86-87: The list of species complexes reads awkwardly here as the conjunction
"and" is incorrectly placed. These lines could be rephrased as follows: "For clarity,
non-geniculate species will be referred to as Phymatolithopsis and Pneophyllum,
while geniculate species will be referred to as Corallina/Arthrocardia fine and
Corallina/Arthrocardia robust."

**Response:** We changed the sentence as suggested.**

• Line 96: Were the irradiances related to minimum and maximum values at the site, why were these specific intervals chosen? This does not seem to be detailed in Krieger et al. (2023) beyond indicating that these levels were observed at the site.

**Response:** The chosen values approximate minimum summer irradiances, which are ecologically relevant as such low-light conditions often dominate under the canopy. Interval selection was partly arbitrary but guided by logistical feasibility, ensuring non-overlapping treatments that were expected to elicit measurable physiological responses.

We added to the text: "The chosen values approximate minimum summer irradiances, which are ecologically relevant as such low-light conditions often dominate under the canopy"

• Lines 98-99: It would be clearer to simply write that "eight header tanks each supplied six different experimental tanks..." The way it is currently written, on first read, the sentence suggests that header tanks only supplied six tanks in total.

**Response:** We changed the sentence as suggested.

• Lines 130-135: Based on the way line 134 defines  $\delta^{11}B_{sw}$ , presumably the equation should show  $\delta^{11}B_{sw}$  instead of  $\delta^{11}B_{seawater}$ . "a" is also not defined as the equilibrium isotopic fractionation factor.

**Response:** We modified  $\delta^{11}B_{sw}$  to  $\delta^{11}B_{seawater}$  within the text to match the equation. We added "[...] and  $\alpha$  representing the fractionation factor and  $\epsilon$  representing the boron isotopic fractionation between boric acid and borate ion (27.2 ‰, Klochko et al., 2006)."

• Line 140: Phrasing reads awkwardly, may want to change to "best described the data".

**Response:** We changed the sentence as suggested.

• Lines 149-150: What was the purpose of the Mantel test, did it assist in interpreting results or was it simply for data presentation? This section does not discuss how it was used, only describes the general definition of Mantel tests.

Response: We are not actually presenting Mantel test but correlation matrices, this has been changed within the text. The idea behind those correlations was mainly to present the data and visually see the significant relationship we also observed based on the different models used to fit the data. We removed the figures from the main text. The paragraph now reads: "Correlation matrices are a statistical method that evaluates the correlation between multiple parameters and allows representation of complex datasets. Correlation matrices were performed using R for each complex and are presented in Figs. S4 and S5. These correlation matrices were used to visually present the data and support interpretation from regression models and other statistical methods used in this paper."

• Lines 150-154: Keep to a single convention when referring to figures, either Fig./Figs. or Figure/Figures. Variation occurs throughout the text when referencing figures.

**Response:** We checked and changed through the text to meet Biogeosciences format.

**Results:**

General comments: The results are overall well-communicated with specific differences between morphologies highlighted as well as across irradiance treatments. The major concern in this section is apparent contradictions in Section 3.7 to other sections in the results and the PCA figure. I recommend reviewing this section for accuracy and adjusting to align with the other results sections.

Specific recommendations:

• Lines 173 & 211: Ensure consistency with how all complexes are referred to (i.e., Corallina/Arthrocardia fine is repeatedly referred to as Corallina which can be confusing to the reader and require re-referencing figures/tables multiple times).

**Response:** We used *Corallina for Corallina/Arthrocardia* fine and *Arthrocardia for Corallina/Arthrocardia* robust in the first draft of the manuscript.

We now changed for Corallina/Arthrocardia fine.

• Line 230: Mantel test results are not discussed at all, are they relevant to include? If the test was conducted for data presentation alone, a short summary of relevant correlations could be included at the end of the section (i.e., switch 3.6 and 3.7) or it could be incorporated into the PCA section of the results. If the the authors agree that this would be a redundancy, the section could be removed altogether, and the Mantel test figures could be moved to the supplemental materials or to the discussion as a summary figure.

**Response:** The correlation matrices were provided to have a visual representation of the data because the dataset is complex. However, we agree that they do not serve as the main interpretation in our paper but they allow complex-specific comparison of the different parameters. We fused sections 3.6 and 3.7 and transferred the correlation matrices to the supplemental information.

• Lines 234-238: If interpretations of relationship in the PCA are made based solely on angles of vectors as indicated by the reference to Figure 4 at the end of the sentence, irradiance and net calcification show an obtuse angle, indicating a minor negative correlation between irradiance and net calcification contrary to previous interpretations of results and what has been written here.  $\delta^{11}$ B and  $F_v/F_m$  seem to show a similar correlation in magnitude and direction to  $\delta^{13}$ C and net calcification (i.e., indicating that  $\delta^{11}$ B and  $F_v/F_m$  correlation may not be minor if that is the case for net calcification and  $\delta^{13}$ Cmineral). Either specific references to correlation coefficients should be made to address the mismatch between the biplot and section 3.7 or the sentence in Lines 237-238 needs to be amended.

**Response:** Thank you for pointing this out. The minor correlation between irradiance and calcification was removed from the text. The paragraph now reads:" Vectors present a positive relationship between ETRmax and irradiance, a negative relationship between net calcification and  $\delta^{11}$ B, positive relationships between net calcification and  $\delta^{13}$ Cmineral and between  $\delta^{11}$ B and Fv/Fm. (Fig. 4 and S3)."

• Lines 239-241: These results appear to relay the exact opposite of what is indicated in section 3.2 (Lines 182-186) and section 3.3 (Lines 197-200). I would assume that this is an error, please amend to reflect the correct results (i.e., geniculate and non-geniculate should be switched to indicate that non-geniculate show higher net

calcification, higher  $\delta^{13}C_{mineral}$ , and lower  $\delta^{11}B$ , while geniculate coralline algae show lower net calcification, lower  $\delta^{13}C_{mineral}$ , and higher  $\delta^{11}B$ .

**Response:** Thank you for noticing, this was a syntax error, we addressed the issue now.

**Discussion:**

General comments: The discussion is generally well-written, particularly sections 4.5 and 4.6. The main recommendations are to provide some clarification on certain claims and references to ensure that they are applicable. Otherwise, comments include minor corrections for grammar, flow and accuracy of statements.

Specific recommendations:

• Line 245: Section title should likely be Carbon isotopes ( $\delta^{13}$ C) as trace element discussion occurs towards the end of the discussion section.

**Response:** The title originally was: "Impact of irradiance is observed on  $\delta^{13}C_{mineral}$  and  $\delta^{13}C_{tissue}$ ", we changed it back.

• Line 247: Is this meant to say  $\delta^{13}$ C in both instances rather than  $^{13}$ C?

Response: We changed it.

• Lines 252-255: Based on the results section and relationships shown in figure fS1 (i.e.,  $\delta^{13}C_{organic}$  results) for geniculate coralline algae,  $\delta^{13}C_{tissue}$  and irradiance do not show a positive relationship. Only non-geniculate complexes show significant relationships, one of which is a positive, linear correlation. Therefore, it is unclear whether the second point in this paragraph can be inferred or supported by the given results. Please review and adjust section for accuracy or clarity of communication.

**Response:** We also added result from ANOVA, the paragraph now reads "The positive relationships between  $\delta 13$ Cmineral and irradiance in three out of four complexes and the significant effect of irradiance on  $\delta 13$ Cmineral (i.e. Corallina/Arthrocardia fine and Phymatolithopsis complex) and  $\delta 13$ Ctissue (i.e. Pneophyllum complex and Phymatolithopsis complex) (p < 0.05, ANOVA), highlights: [...]"

• Line 263: Authors may consider rephrasing here to indicate that photosynthesis impacts the  $\delta^{13}$ C of the available DIC for calcification. "Enhancing" suggests that photosynthesis increases rate of net calcification (e.g., through increase of pH).

Response: The paragraph now reads "There are clear differences in  $\delta^{13}C_{mineral}$  signatures between non-geniculate and geniculate complexes. Non-geniculate complexes Pneophyllum complex and Phymatolithopsis complex are fast calcifiers that have enriched  $\delta^{13}C_{mineral}$  and a strong response to increased irradiance. Geniculate complexes Corallina/Arthrocardia fine and Corallina/Arthrocardia robust present lower net calcification and lower  $\delta^{13}C_{mineral}$ . Photosynthesis can increase the  $\delta^{13}C$  of the DIC pool available for

calcification, the differences observed between morphotypes in  $\delta^{13}C_{\text{mineral}}$  and net calcification are then in line with a positive effect of photosynthesis on net calcification (Fig. 5C). "

• Lines 273-276: Please review if Mao et al. (2024) is relevant to this case. The reference seems to indicate that  $CO_2$  produced from calcification is recycled for photosynthesis and not vice versa. However, as it is written here, the explanation suggests that carbon used for photosynthesis is recycled internally for calcification thereby affecting  $\delta^{13}C_{\text{mineral}}$ .  $HCO_3^-$  is actively pumped into cells for calcification and photosynthesis.  $CO_2$  produced by calcification or respiration may be recycled for photosynthesis, and products of photosynthesis like ATP are used for calcification, however it would not follow that carbon from photosynthesis would be directly recycled for calcification. Re-wording of this section or clarification may be necessary. It may be possible for  $HCO_3^-$  released from respiration to be recycled for calcification, thereby reducing  $\delta^{13}C_{\text{mineral}}$ , and as inputs for respiration are derived from photosynthesis,  $\delta^{13}C$  of DIC available for calcification could be indirectly affected by photosynthesis.

**Response:** Thank you for noticing, this now reads: ". Mao et al. (2024) established a carbon budget based on radiogenic-isotopes and highlighted that up to 40% of the carbon released during calcification was recycled internally. While carbon fixed during photosynthesis is not directly recycled into calcification,  $CO_2$  released during respiration may contribute to calcification, potentially lowering  $\delta^{13}C_{\text{mineral}}$ . Because respiratory inputs are derived from photosynthetically fixed carbon,  $\delta^{13}C$  of the DIC pool available for calcification could be indirectly influenced by photosynthesis. We anticipate [...]"

• Lines 279-280: Rather than "allows us", it may be more accurate to write something to the effect of: "Our results show that the geochemical signatures of the mineral are impacted by changing irradiances indicating potential changes in pHCF, which we analyzed by boron isotope proxy."

**Response:** The proposed sentence slightly changed the original sense, we revised: "Our results show that the geochemical signatures of the mineral are impacted by changing irradiances thereby enabling the investigation of potential changes in pHCF constrained by boron isotopes.".

• Line 376: "Few differences", should likely be changed to "a few differences," or simply "differences between morphologies...".

**Response:** We changed for simply "differences".

• Line 415: I would restructure this section as encrusting species are much more commonly and successfully used for paleoenvironmental reconstructions than rhodoliths in coralline algae literature. Records produced from encrusting individuals are less impacted by differential light exposure since they are anchored to an

unmoving substrate unlike free-living rhodoliths where a face of the organism is always buried in sediment.

Response: Thank you for this comment, this section now reads: "Carbonate structures produced by coralline algae (e.g., rhodoliths, crusts) can be used as archives for paleoreconstruction (MacDonald et al., 2024). The main geochemical differences in our study are observed between the different morphologies of coralline red algae. Nevertheless, non-geniculate (i.e., encrusting) species are much more commonly used for paleoenvironmental reconstructions, we will then focus on the non-geniculate complexes for the rest of this section.

As we observed,  $\delta^{11}$ B-derived pHCF is not impacted by light at the complex levels which does not produce additional complexity for the use of the proxy. Anagnostou et al. (2019) presented a robust calibration of the  $\delta^{11}$ B proxy based on culture experiments on a high-latitude crustose coralline red algae *Clathromorphum compactum*. As the carbonate archives usually are produced by a mix of species, a complex-specific response to ocean acidification and the strong control they exert on their calcification fluid could be a limitation of the proxy, but our findings suggest  $\delta^{11}$ B should be at least insensitive to light levels. This is especially true because encrusting species being anchored to the substrate should be less impacted by differential light exposure. Nevertheless, with the increasing availability in species-specific geochemical data, a rigorous approach may involve using DNA-based identification within the core to calibrate geochemical records."

• Line 420: Adjust this section of the sentence to make grammatical sense: "which does not produce additional complexity the use of the proxy."

**Response:** We changed for "for the use of the proxy".

Line 425: It may also be relevant to include that a multi-proxy approach could be applied to proxies like Mg/Ca that are affected by multiple variables. Additionally, light availability would likely affect species adapted to different latitudes and depths uniquely in addition to differences in effects by morphology, so it would be beneficial to indicate that the results could possibly apply to other mid-latitude species but not all coralline algae (e.g., Arctic species are adapted to much lower light conditions where it has been suggested that stored photosynthates can be used to support calcification during winter months as indicated by Adey et al. (2013) and Gould et al. (2022)).

**Response:** This section now reads: "Our results on mid-latitude low-light adapted species show that light does not impair the application of the  $\delta^{11}B$  and Li/Mg proxies."

Coralline red algae species are adapted to environments where light availability can vary (e.g. latitude, depth). While the results of this study may be applicable to mid-latitude species, it might not be transferable to coralline algae from other latitudes, for example, it has been shown that Arctic species rely on stored photosynthates to support winter

calcification (Adey et al., 2019; Gould et al., 2022) which could influence the geochemical parameters."

**Conclusion:**

General comments: The conclusion provides an excellent summary of the research and pertinent results as well as interpretations. The only recommendation would be to acknowledge that as study results may be species-specific and morphology specific, they could be cautiously generalized to mid-latitude species but additional replication of the study is necessary for species adapted to different light regimes.

**Specific recommendations:**

• Line 471: Additional studies should also be repeated with different coralline algal species that experience different irradiance regimes and environments (i.e., are there differences between algal species that are adapted to living at greater depths and higher/lower latitudes with lower access to light).

Response: The end of the conclusion now reads: "This study demonstrates variability in responses of coralline red algae under irradiance and highlights distinct biomineralization mechanisms between branching (geniculate) and encrusting (non-geniculate) mid-latitude low-light adapted complexes. Photosynthesis impacts the availability and source of DICCF which has implications on calcification. In the perspective of calcification, plasticity on DIC sources is determinant for acclimation of coralline red algae. Further research should be done on coralline algal species that experience different irradiance regimes and environments (e.g. latitude, depth). Additional study on the joint effect of ocean acidification and changing irradiance might provide some interesting dynamics and will be needed to understand the full implications of future global changes and associated perturbations on the coralline algae communities and dependent ecosystems. "

**Figures:**

• Figures 4, 6, 7, 8: Figures require more detailed figure captions, including drawing reader attention to pertinent results accompanied by applicable statistics. All figures should follow the same format in describing sub-figures in the caption.

**Response**: Figure 4 now only includes the PCA. We added a more detailed caption to the figures.

• Figures 5, 7, S3 & S5: Color schemes should be accessible and consistent across figures (e.g., Figure 8 is not accessible to those with blue-yellow color blindness, Figures 5, 7, and S5 are not accessible to those with red-green color blindness).

**Response:** Originally the color scheme was checked but we see it did not fit all color blindness, color scheme has been changed to #E69F00,#56B4E9, #009E73,#CC79A7.

Figures 5 & 7: The changes between irradiance are quite difficult to distinguish with
the size of the data points. Either the size must be increased, or figures should be
separated. Alternatively, the four average data points could be colored to represent
irradiances and individual data points in the background could be eliminated if
sample size was indicated in the legend or figure caption.

**Response:** Individual datapoints were increased. We left the other averages in black to avoid making the figure overwhelming.

• Figure 5: Are each of the four black-filled and black-outlined geometric shapes representing individual coralline algae averages per species, morphology type and irradiance as shown in previous figures while the smaller data points are all individual measurements taken at each irradiance as in Figure 3? Please include this in the figure caption to clarify in more detail. Observing trends based on irradiance as described in body of paper is difficult in these figures, for example in Figure 5A the highest irradiance for Pneophyllum showes lower  $\delta^{13}C_{\text{mineral}}$  than at the second highest irradiance. Does this indicate that at 2.3 mol photon/m²/day the point at which photochemical quenching is at its maximum has been exceeded? If so, this should be noted, as it appears to be inconsistent with the claim in the discussion.

**Response:** Figure 5 presents crossplots of the geochemical analyses with physiological data. "Averages are calculated based on this study for geochemical parameters and from the full dataset in Krieger et al. (2023). Individual paired data are also shown to maximize the information displayed, color scheme corresponds to the different irradiances. ". These figures aim to evaluate the relationship between the geochemical parameters and other key physiological parameters (e.g. gross photosynthesis and net calcification). The effect of irradiance is studied through the statistical test and the parameters vs irradiance. Those figures highlight the different clusters between morphotypes.

• Figures S1& S2: Y-axis labels are missing for these figures. Ensure to be consistent with the inclusion of R2 and p-values across the figures. At minimum, both should be included for significant results if not all.

**Response:** We added the X-axis labels to those figures. Only R2 is provided for non-linear regressions. We added the R2 to the linear regressions.

[revised manuscript text omitted]
 (mgCaCO3/cm²/day), D. δ11B (‰) and Net Calcification (mgCaCO3/cm²/day), E. δ13Cmineral (‰)
- 745 and gross photosynthesis ( $\mu g O_2/cm/h$ ) and F.  $\delta^{11}B$  (‰) and gross photosynthesis ( $\mu g O_2/cm/h$ ).

746

- Figure 6: pHCF calculated from  $\delta^{11}$ B against irradiance for the four complexes, A. Corallina/Arthrocardia robust, B.
- 748 Corallina/Arthrocardia fine, C. Pneophyllum complex, D. Phymatolithopsis complex. Average values per treatment are
- presented with 2 SD error bars. Individual datapoints are also presented to assess variability within treatment.

750

- 751 **Figure 7:** Multi-panel plots showing crossplots of pHCF, A. net calcification (mgCaCO3/cm2/day), B. gross photosynthesis
- 752 (μgO2/cm/h), C. residual full-width-half-maximum, FWHM, D. δ13Cmineral (‰) and E. Mg/Ca (mmol/mol). Large symbols
- show averages derived from full dataset from Krieger et al. (2023) while small colored symbols show individual paired data
- and irradiance level to display maximum information. Error bars are shown as 2 SD.

[revised manuscript text omitted]

**Supplemental Figures**

Figure S1: Data and significant models (black line) for the geochemical parameters measured and used in this study.

Figure S2: Data and significant models (black line) for the physiological parameters from Krieger et al. (2023) and used in this study.

**Figure S3:** Principal component analyses for (a) the relevant geochemical and physiological parameters used in this study and (b) elemental ratios and physiological parameters.

**Figure S4:** (a-d) Correlation matrices providing pairwise correlations between geochemical, and photo physiological data for a. *Corallina/Arthrocardia* "robust", b. Corallina/Arthrocardia "fine", c. *Pneophyllum* complex and d. *Phymatolithopsis* complex.

Figure S5: Correlation matrices for (a) the geniculate complexes and (b) the non-geniculate complexes.

**Figure S6:** Cross-plots of  $\delta^{13}$ Cmineral and  $\delta^{11}$ B for other photo-physiological parameters, (a) and (b) Gross photosynthesis, (c) and (d) for ETRmax, (e) and (f) for Chl a.

**Figure S7:** Cross-plots of B/Ca with (a)  $\delta^{11}$ B and (b) Chl a.

**Supplemental Tables**

Table S1: Geochemical and physiological data.

Table S2: Comparison of linear and quadratic models based on AIC for the geochemical parameters measured in this study.

**Table S3:** Comparison of linear and quadratic models based on AIC for the physiological parameters published in Krieger et al., (2023).

**Table S4:** ANOVA testing geochemical and physiological data against changing irradiance.

**Table S5:** T-test for parameters presenting significant ANOVA with changing irradiance (from Table S4).

**Table S6:** ANOVA testing geochemical and physiological data between complexes.

**Table S7:** T-test for parameters presenting significant ANOVA when testing for differences between complexes.

**Table S8:**  $\delta^{11}$ B of NIST 8301, JCp-1 and seawater measured in this study.

**Physiological Data PCA**

**Trace Element Data PCA**

**Correlation matrices**

**Correlation matrices**

**Geniculate complexes**

**Non-geniculate complexes**

**Non-geniculate complexes**

☐ Pneophyllum complex

△ *Phymatolithopsis* complex

**Geniculate complexes**

- ◆ Corallina/Arthrocardia "robust"
- Corallina/Arthrocardia "fine"

Irradiance 0.6

Irradiance 1.2

Irradiance 1.8
Irradiance 2.3